# *Arabidopsis* R1R2R3-Myb proteins are essential for inhibiting cell division in response to DNA damage

Poyu Chen[1], Hirotomo Takatsuka[1], Naoki Takahashi[1], Rie Kurata[1], Yoichiro Fukao[2], Kosuke Kobayashi[3], Masaki Ito[3,4] & Masaaki Umeda[1,5]

Inhibition of cell division is an active response to DNA damage that enables cells to maintain genome integrity. However, how DNA damage arrests the plant cell cycle is largely unknown. Here, we show that the repressor-type R1R2R3-Myb transcription factors (Rep-MYBs), which suppress G2/M-specific genes, are required to inhibit cell division in response to DNA damage. Knockout mutants are resistant to agents that cause DNA double-strand breaks and replication stress. Cyclin-dependent kinases (CDKs) can phosphorylate Rep-MYBs in vitro and are involved in their proteasomal degradation. DNA damage reduces CDK activities and causes accumulation of Rep-MYBs and cytological changes consistent with cell cycle arrest. Our results suggest that CDK suppressors such as CDK inhibitors are not sufficient to arrest the cell cycle in response to DNA damage but that Rep-MYB-dependent repression of G2/M-specific genes is crucial, indicating an essential function for Rep-MYBs in the DNA damage response.

[1] Graduate School of Biological Sciences, Nara Institute of Science and Technology, Takayama 8916-5, Ikoma, Nara 630-0192, Japan. [2] Department of Bioinformatics, Ritsumeikan University, Kusatsu, Shiga 525-8577, Japan. [3] Graduate School of Bioagricultural Sciences, Nagoya University, Chikusa, Nagoya 464-8601, Japan. [4] JST, CREST, Chikusa, Nagoya 464-8601, Japan. [5] JST, CREST, Takayama 8916-5, Ikoma, Nara 630-0192, Japan. Correspondence and requests for materials should be addressed to M.U. (email: mumeda@bs.naist.jp)

Genotoxic stress threatens genome integrity and is unavoidable for every organism. Various kinds of environmental stress as well as cellular processes (e.g., DNA replication, metabolism) can cause DNA damage; in plants, high boron or aluminium levels in soil, pathogen attack and stress-induced reactive oxygen species are known to damage genomic DNA[1–4]. To prevent the transmission of damaged DNA to daughter cells, cell cycle progression is delayed or arrested before the cell enters DNA replication or mitosis. In animals, the DNA damage checkpoint is triggered by the sensor kinases ATM (ataxia telangiectasia mutated) and ATR (ATM-related and Rad3-related); ATM senses DNA double-strand breaks (DSBs), while ATR is primarily activated by replication stress and single-strand breaks (SSBs)[5, 6]. Downstream of these kinases, checkpoint-1 and checkpoint-2 kinases (CHK1 and CHK2) phosphorylate and stabilize the transcription factor p53, which then induces expression of the CDK inhibitor p21 and reduces the activity of cyclin-dependent kinases (CDKs)[7, 8]. Plants also possess functional ATM and ATR[9, 10], but these downstream regulators are all missing in plants; instead, suppressor of gamma response 1 (SOG1) has been identified in *Arabidopsis* as a plant-specific transcription factor that controls the DNA damage response[11]. Recent reports have shown that ATM and ATR directly phosphorylate and activate SOG1[12, 13].

DSBs inhibit plant cell division, but also induce an early transition from cell division to endoreplication, a repeated cycle of DNA replication without mitosis or cytokinesis[14]. Endoreplication is triggered by blocking the G2-to-M progression; indeed, in *Arabidopsis* cultured cells, DSBs arrest the cell cycle at G2 before provoking endoreplication[14]. Another typical response to DSBs is cell death around the stem cell region of shoot and root meristems[15, 16]. Both endoreplication and cell death occur in an ATM/ATR- and SOG1-dependent manner, implying a programmed response to DNA damage. We previously reported that DSBs increase the expression of CDK suppressors, such as CDK inhibitors, and downregulate cyclin A (CYCA) and B (CYCB) genes, which are essential for CDK activation at G2/M[14]. However, it remains unknown whether such a dramatic change in expression of cell cycle regulators is a cause or a consequence of G2 arrest.

In plants, many G2/M-specific genes, including those for mitotic cyclins, kinesin-like proteins and the cytokinesis-specific syntaxin KNOLLE, are controlled by three Myb repeat-containing transcription factors, called R1R2R3-Myb (MYB3R)[17]. The *Arabidopsis* genome has five *MYB3R* genes, *MYB3R1* to *MYB3R5*. MYB3R1 and MYB3R4 act as transcriptional activators, while MYB3R3 and MYB3R5 are transcriptional repressors[18–20]. MYB3R1 exhibits a repressor function as well; therefore, MYB3R1 and MYB3R4 are known as the activator-type (Act-MYBs), and MYB3R1, MYB3R3 and MYB3R5 as the repressor-type (Rep-MYBs)[20]. Although both Act- and Rep-MYBs bind to the cis-acting MSA element on target gene promoters, Rep-MYBs repress transcription along the cell cycle except G2/M in proliferating cells, and suppress G2/M-specific genes in post-mitotic cells during organ development[20].

In this study, we show that Rep-MYBs play a crucial role in inhibiting cell division under DNA damage conditions. We propose a model whereby a reduction in CDK activity leads to accumulation of Rep-MYBs, which in turn suppresses mitotic genes, representing an elaborate mechanism for stress-induced cell cycle arrest.

## Results

**myb3r3 and myb3r5 mutants are tolerant to genotoxic stress.** Since our previous observation demonstrated that DSBs trigger

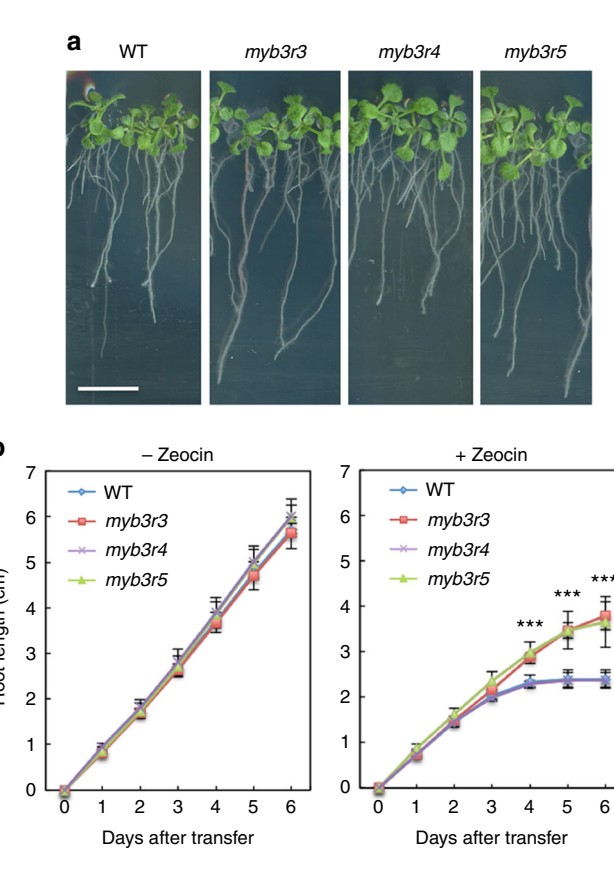

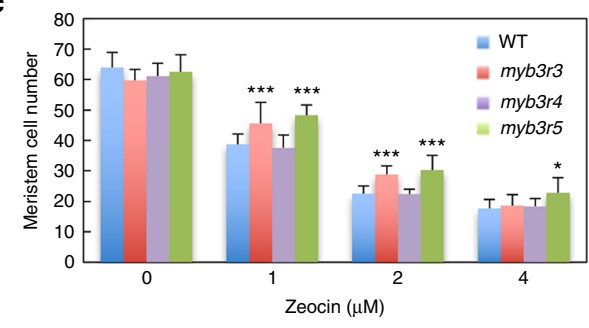

**Fig. 1** Zeocin response of *myb3r* mutants. **a** Seedlings of WT and *myb3r* mutants. Five-day-old seedlings were transferred to medium with 2 μM zeocin and grown for 6 days. *Scale bar*, 1 cm. **b** Root growth of WT and *myb3r* mutants. Five-day-old seedlings were transferred to medium with or without 2 μM zeocin, and root length was measured for 6 days. **c** Meristem size of WT and *myb3r* mutants. Five-day-old seedlings were transferred to medium with different concentrations of zeocin, and cortex cell number between the quiescent centre and the first elongated cell was counted after 24 h. Data are presented as mean ± SD (n = 30). Significant differences from WT were determined by Student's *t*-test: *$P < 0.05$; ***$P < 0.001$

G2 arrest in *Arabidopsis* cultured cells[14], we first examined the involvement of MYB3Rs in the DNA damage response. Among the five MYB3Rs, MYB3R2 is only distantly related to the other MYB3Rs, and MYB3R1 acts as both an activator and a repressor, and exhibits a supplementary function; indeed, in the *myb3r1* knockout mutant, G2/M-specific genes are expressed at the same levels as they are in wild-type (WT)[20]. Therefore, we excluded these two MYB3Rs from further analyses, and observed T-DNA insertion mutants of *MYB3R3*, *MYB3R4* and *MYB3R5*. Seedlings grown on Murashige and Skoog (MS) medium for 5 days were transferred to medium with or without 2 μM zeocin, an inducer of DSBs[21], and grown for a further 6 days (Fig. 1a). *myb3r3-1* and

*myb3r5-1* (hereafter called *myb3r3* and *myb3r5*, respectively), which completely lack normal transcripts[20], displayed higher tolerance to zeocin than WT (Fig. 1b). Faster root growth was also observed at lower (1 μM) and higher (4 μM) zeocin concentrations (Supplementary Fig. 1). Similar results were obtained with two other alleles, *myb3r3-2* and *myb3r5-2*, as shown in Supplementary Fig. 2. The *myb3r3 myb3r5* double mutant was tolerant to zeocin to an extent similar to each single mutant (Supplementary Fig. 3a), implying that both *MYB3R3* and *MYB3R5* are essential for root growth arrest. Zeocin tolerance was comparable between *myb3r3*, *sog1-1* and the *myb3r3 sog1-1* double mutant (Supplementary Fig. 3b), suggesting that Rep-MYBs function in SOG1-mediated inhibition of root growth.

*myb3r4-2* (hereafter called *myb3r4*) has a T-DNA insertion 35 bp upstream of the start codon, and the *MYB3R4* transcript level decreases to less than 3% of the WT level[18]. In this mutant, root growth was inhibited by zeocin to the same extent as in WT (Fig. 1b). To examine the possibility that residual *MYB3R4* transcripts function in *myb3r4*, we used another allele, *myb3r4-3*, which carries two copies of T-DNA insertions and produces no *MYB3R4* transcripts (Supplementary Fig. 4). Our data showed that roots exhibited the same sensitivity to zeocin as that in WT and *myb3r4-2* (Supplementary Fig. 2), indicating that *MYB3R4* is not involved in controlling root growth in response to DSBs.

Observation of the root tip showed that increasing zeocin concentrations resulted in weaker reduction of the meristem cell number in *myb3r3* and *myb3r5*, but not in *myb3r4*, compared to WT (Fig. 1c and Supplementary Fig. 5). Moreover, propidium iodide-stained dead cells were observed in stem cells and stele precursor cells of WT and *myb3r4* in the presence of 1 μM zeocin, while a higher concentration was required for *myb3r3* and *myb3r5* (Supplementary Fig. 5). These results suggest that Rep-MYBs participate in the inhibition of cell division and induction of stem cell death in response to DSBs.

Next, we tested the response of *myb3r3* and *myb3r5* to other DSB-inducing agents, namely bleomycin, bleocin, gamma irradiation and boron overload. These mutants were significantly more tolerant to all four treatments than WT (Supplementary Fig. 6). We then examined the response to hydroxyurea (HU) and aphidicolin, which inhibit DNA replication by decreasing the dNTP pool and by inhibiting DNA polymerase α, respectively[22, 23]. Although HU had a milder effect than aphidicolin, both treatments revealed tolerance in *myb3r3* and *myb3r5* (Supplementary Fig. 6). These results suggest that Rep-MYBs function in the response to replication stress as well as to DSBs.

To investigate whether Rep-MYBs are also involved in the control of cell division in the shoot apex, we irradiated *Arabidopsis* seeds with 200 or 400 Gy of gamma rays and grew the seedlings for 10 days in MS medium, followed by counting the number of true leaves. *sog1-1* and *lig4*, the latter of which is defective in non-homologous end-joining, exhibited lower and higher sensitivity to gamma irradiation, respectively, as reported previously (Supplementary Fig. 7)[11, 24]. *myb3r3*, *myb3r5* and the *myb3r3 myb3r5* double mutant produced more true leaves than WT at 400 Gy (Supplementary Fig. 7), suggesting that Rep-MYBs also function in DNA damage-induced inhibition of cell division in the shoot apex.

**Rep-MYB proteins accumulate highly under DNA stress**. To examine whether MYB3R expression responds to DNA damage, we conducted quantitative real-time PCR (qRT-PCR) using RNA from root tips. The transcript level of *MYB3R4* decreased by about 70% within 1 h of zeocin treatment, while those of *MYB3R3* and *MYB3R5* did not change during the 12-h treatment

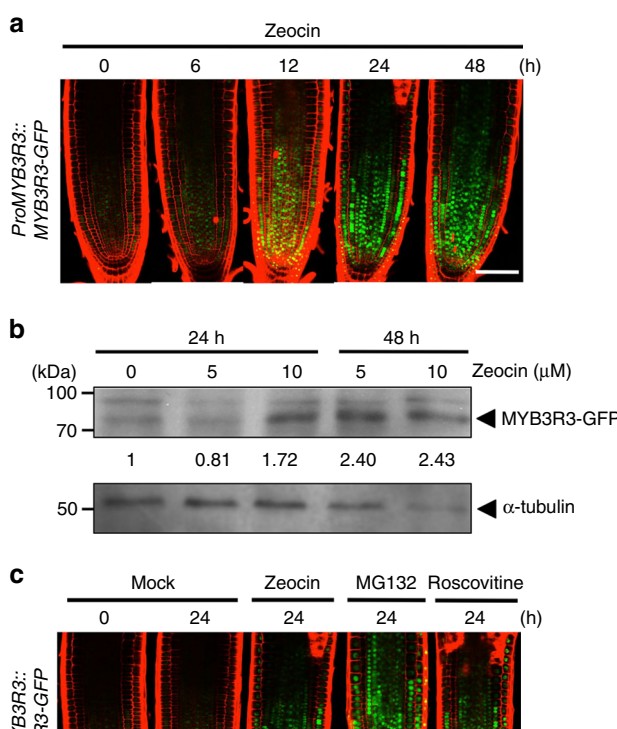

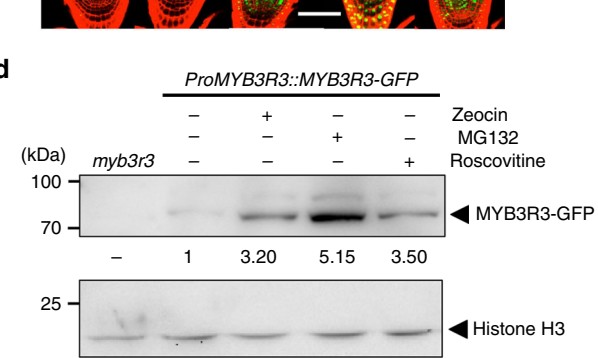

**Fig. 2** MYB3R3 accumulation in the root tip. **a**, **c** Confocal microscopy images of root tips. Five-day-old *myb3r3* seedlings harbouring *ProMYB3R3::MYB3R3-GFP* were treated with 2 μM zeocin for the indicated times **a**, or with 2 μM zeocin, 50 μM MG132 or 25 μM roscovitine for 24 h **c**. *Scale bars*, 100 μm. **b**, **d** Protein level of MYB3R3-GFP in roots. Ten-day-old *myb3r3* seedlings harbouring *ProMYB3R3::MYB3R3-GFP* were treated with 5 or 10 μM zeocin for 24 or 48 h **b**, or with 2 μM zeocin, 50 μM MG132 or 25 μM roscovitine for 24 h **d**. Forty (**b**) or 20 μg (**d**) of total protein extracted from roots were subjected to immunoblotting using antibodies against GFP, α-tubulin or histone H3. Protein extract from the *myb3r3* mutant was used as a control (**d**). Relative levels of MYB3R3-GFP are expressed as the fold change, normalized with respect to the band of α-tubulin or histone H3

(Supplementary Fig. 8). This result agrees with data obtained from zeocin-treated cultured cells[14].

Since Rep-MYBs did not respond to zeocin at the mRNA level, we next observed protein accumulation of MYB3R3 using *ProMYB3R3::MYB3R3-GFP*, which carries the 1.3-kb promoter and the protein-coding region of *MYB3R3*. This reporter gene can complement the *myb3r3* mutation, indicating that the MYB3R3-GFP fusion protein is functional[20]. GFP signals were very faint in the absence of zeocin, but became pronounced in the meristem after 12 h of zeocin treatment and thereafter (Fig. 2a). Immuno-blot analysis of total protein from roots also showed marked

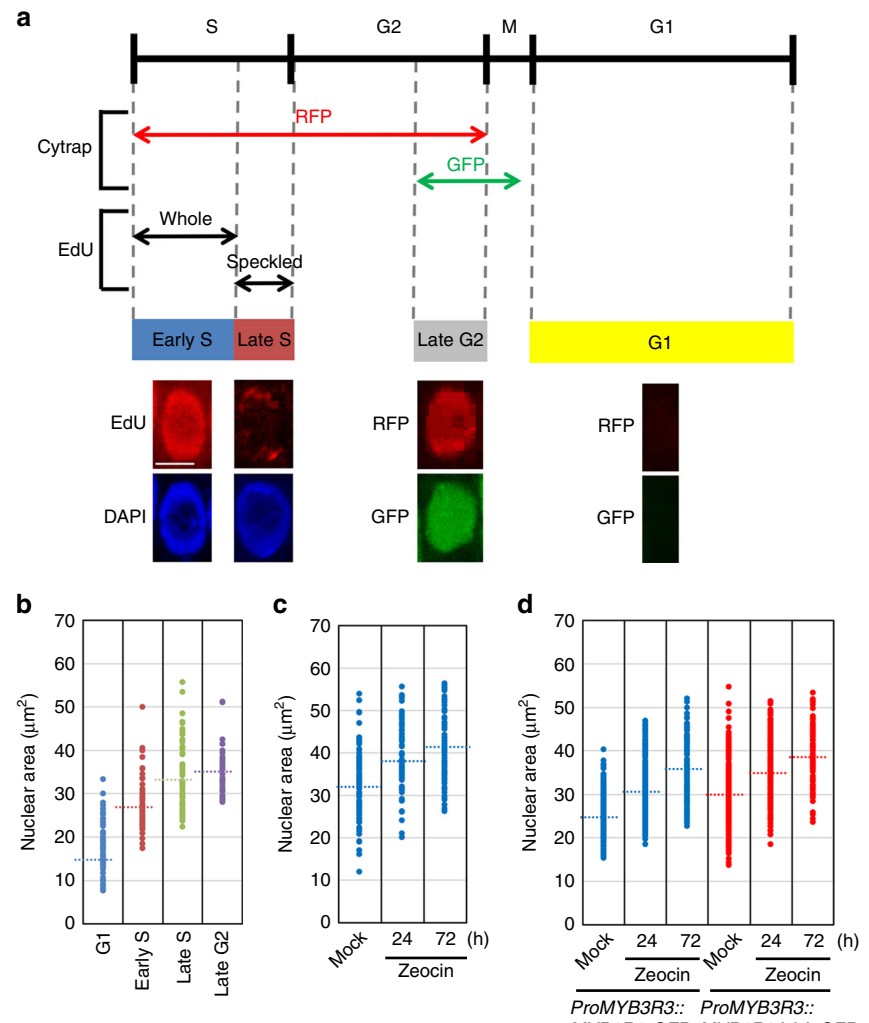

**Fig. 3** Changes in nuclear area upon DNA damage. **a** Assignment of root meristematic cells to distinct cell cycle stages. In the Cytrap marker system, RFP and GFP fluorescence represent cells in S-to-G2 and late G2-to-M (metaphase), respectively. EdU staining of the whole nucleus or speckled signals indicate cells in early S or late S, respectively. Representative patterns of EdU and Cytrap signals are shown for early S, late S, late G2 and G1. *Scale bar*, 10 µm. **b** Nuclear area at distinct cell cycle stages. By using 5-day-old seedlings, cell cycle stages of cells in the root meristem were identified based on Cytrap and EdU signals as shown in **a**. *Dotted lines* indicate the average nuclear area ($n > 68$). **c** Nuclear area of root meristematic cells after treatment with zeocin. Five-day-old WT seedlings were transferred to medium with 2 µM zeocin and grown for 24 or 72 h. *Dotted lines* indicate the average nuclear area ($n > 84$). **d** Nuclear area in the *myb3r3* mutant harbouring *ProMYB3R3::MYB3R3-GFP* or *ProMYB3R3::MYB3R3AAA-GFP*. Five-day-old seedlings were transferred to medium with 2 µM zeocin and grown for 24 or 72 h. Nuclear area of GFP-expressing cells in the root meristem was measured. *Dotted lines* indicate the average nuclear area ($n > 118$)

accumulation of MYB3R3-GFP, dependent on zeocin concentration and treatment time (Fig. 2b). When the proteasome inhibitor MG132 was applied to the reporter line in the absence of zeocin, GFP fluorescence was again enhanced in the root tip, and dramatic accumulation of MYB3R3-GFP was observed by immunoblotting (Fig. 2c, d). Similar patterns of protein accumulation upon zeocin or MG132 treatment were also observed for MYB3R5, albeit to a lesser extent than MYB3R3 (Supplementary Fig. 9). These data suggest that Rep-MYBs are actively degraded via the ubiquitin-proteasome pathway under normal growth conditions, but accumulate to high levels in response to DNA damage.

We also examined protein accumulation of MYB3R4 using *ProMYB3R4::MYB3R4-GFP*. Since the MYB3R4-GFP fusion protein was barely detected by immunoblotting, we quantified GFP fluorescence in the root tip to estimate the protein level. The amount of MYB3R4-GFP was greatly reduced within 6 h of zeocin treatment (Supplementary Fig. 10a, b), correlating with a

rapid decrease in the transcript level (Supplementary Fig. 8). Unlike Rep-MYBs, MG132 treatment did not change the protein level of MYB3R4-GFP under normal growth conditions (Supplementary Fig. 10c, d), suggesting that MYB3R4 is not under the control of proteasomal degradation.

**DSBs induce cell cycle arrest in roots**. Since Rep-MYBs are involved in repression of G2/M-specific genes[20], it is possible that their accumulation causes cell cycle arrest. We therefore investigated cell cycle stages of root meristematic cells under DNA damage conditions. Flow cytometry analysis using root tip cells cannot distinguish mitotic 4C cells from endocycling 4C cells, which increase in response to DSBs[14]. Moreover, the G2/M-marker gene *CYCB1;1* is responsive to DNA damage[25]; thus, we took another approach. Since nuclear DNA content, which increases during progression through S phase, is well correlated with nuclear size[26], we measured the nuclear area of

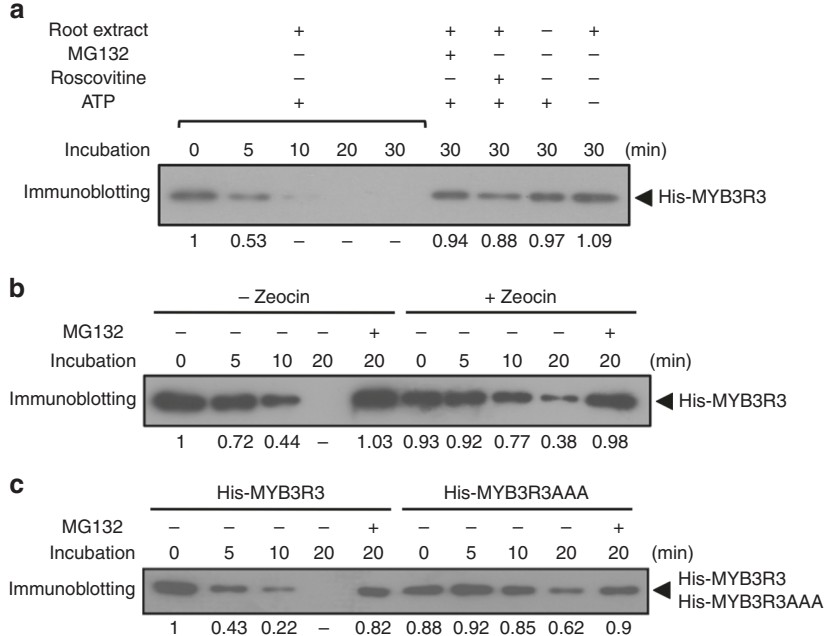

**Fig. 4** In vitro degradation assay of MYB3R3. Recombinant His-MYB3R3 protein was incubated with or without root extracts from 7-day-old WT seedlings, and His-MYB3R3 was immunoblotted with anti-His antibodies. **a** Protein mixtures were incubated in the presence or absence of 50 μM MG132, 25 μM roscovitine and/or 10 mM ATP for the indicated times. **b** Total protein was extracted from 7-day-old WT roots treated with or without 2 μM zeocin for 24 h. Protein mixtures were incubated in the presence or absence of 50 μM MG132 for the indicated times. **c** Recombinant His-MYB3R3 or His-MYB3R3AAA proteins were incubated with root extracts in the presence or absence of 50 μM MG132 for the indicated times. Relative levels of His-MYB3R3 and His-MYB3R3AAA are expressed as the fold change, normalized with respect to the band of recombinant proteins visualized by CBB staining

each meristematic cell to estimate cell cycle stages. A previous report showed that the incorporation of 5-ethynyl-2′-deoxyuridine (EdU) into DNA stains whole nuclei of early S-phase cells, while speckled signals are detected for late S-phase cells due to DNA replication of heterochromatic regions[27]. Thus, we used EdU staining to identify early and late S-phase cells (Fig. 3a). G1- and G2-phase cells were searched for using the dual-colour marker system Cytrap; as previously reported[28], cells exhibiting both RFP and GFP fluorescence were identified as late G2-phase cells, and non-fluorescing cells as G1-phase cells (Fig. 3a). Our measurement of nuclear area at each cell cycle stage showed that it increased in the progression of G1, early S, late S and late G2 (Fig. 3b), reflecting the elevation of DNA content during replication.

When WT seedlings were treated with zeocin, the nuclear area increased significantly (Fig. 3c), suggesting cell cycle arrest during S/G2, but not G1. To further investigate cell cycle progression in the presence of zeocin, roots were treated with EdU for 15 min, and the number of EdU-labelled cells with mitotic figures was then counted. Our results showed that the percentage of EdU-labelled cells among those with mitotic figures was elevated in the absence of zeocin and reached a maximum after 6 h, while such an increase was delayed and suppressed after treatment with zeocin (Supplementary Fig. 11). This result suggests that DSBs inhibit G2 progression in root meristematic cells, thereby preventing their entry into M phase.

To examine whether Rep-MYB accumulation is associated with zeocin-induced cell cycle arrest, we then used the *myb3r3* mutant harbouring *ProMYB3R3::MYB3R3-GFP*. When nuclei of GFP-expressing cells were observed in the absence of zeocin, the distribution of nuclear area was similar to that in early S-phase cells (Fig. 3b, Early S; Fig. 3d, Mock). This suggests that MYB3R3 protein preferentially accumulates around early S phase under normal growth conditions. We found that zeocin treatment increased the nuclear area of GFP-expressing cells in a time-

dependent manner: the average nuclear area increased at 24 h and reached that of late G2-phase cells at 72 h (Fig. 3d). Although measuring the nuclear area alone cannot clearly identify cell cycle phases, this result suggests that zeocin-induced DSBs enhance MYB3R3 accumulation, thereby leading to cell cycle arrest.

**Rep-MYB degradation is dependent on CDK activities**. The next question is how Rep-MYB accumulation is controlled in response to DNA damage. Araki et al.[29] demonstrated that NtmybA2, one of the tobacco Act-MYBs, is phosphorylated and activated by CDK. A transient transactivation assay showed that *Arabidopsis* MYB3R4 is activated by co-expression with *CYCB1*[18]. Therefore, we speculated that Rep-MYBs are also under the control of CDK. To test this possibility, we first treated plants carrying *ProMYB3R3::MYB3R3-GFP* or *ProMYB3R5::MYB3R5-GFP* with the CDK-specific inhibitor roscovitine. We observed a marked increase in GFP signals and protein levels of MYB3R3- and MYB3R5-GFP after 24 h of roscovitine treatment (Fig. 2c, d and Supplementary Fig. 9). The same treatment did not change the transcript levels of *MYB3R3* or *MYB3R5* (Supplementary Fig. 12), indicating that Rep-MYB accumulation upon roscovitine treatment is controlled at the protein level. These results suggest that CDK activity is required for Rep-MYB degradation under normal growth conditions. We also examined the transcript and protein levels of MYB3R4 and found that they were elevated upon roscovitine treatment for some unknown reason (Supplementary Figs. 10c, d and 12). As described above, *MYB3R4* expression is rapidly reduced by zeocin treatment, although DNA damage reduces CDK activity. This indicates the existence of a mechanism to actively downregulate *MYB3R4* expression in response to DNA damage.

We then examined whether protein stability of MYB3R3 is affected by CDK activity or DNA damage by performing an in vitro assay according to a previously described method[30]. Histidine (His)-tagged MYB3R3 produced in *Escherichia coli* was

incubated with total protein extracted from WT roots, and the remaining amount of His-MYB3R3 was estimated by immunoblotting using anti-His antibody. A reduction of His-MYB3R3 was observed after a 5-min incubation, and almost no protein was detected after 20 min (Fig. 4a). However, when MG132 or roscovitine was included in the reaction, His-MYB3R3 remained detectable after a 30-min incubation (Fig. 4a). When ATP was omitted from the reaction, His-MYB3R3 degradation was suppressed (Fig. 4a), suggesting that MYB3R3 phosphorylation and/or ATP-dependent ubiquitination are impaired, thus preventing proteasomal degradation. Next, we used protein extracts from zeocin-treated roots; His-MYB3R3 was still detected after a 20-min incubation, indicating slower degradation than in the non-treated control (Fig. 4b). This and the above results of GFP reporter lines suggest that proteasome-mediated degradation of MYB3R3 is dependent on CDK activities and is suppressed by DNA damage.

**CDK phosphorylates the C-terminal region of MYB3R3**. To examine whether Rep-MYBs are directly phosphorylated by CDK, we conducted an in vitro kinase assay. In plants, two types of CDKs control cell cycle progression: CDKA is an orthologue of mammalian CDK1 and yeast Cdc2/Cdc28, and CDKB is a plant-specific type with two subfamilies, CDKB1 and CDKB2[31]. Each CDK was immunoprecipitated from *Arabidopsis* cultured cells using specific antibodies[32–34], and subjected to kinase assay. As shown in Fig. 5a, all three CDKs phosphorylated His-MYB3R3 and His-MYB3R5, but not His-GFP. CDKA exhibited a higher kinase activity than CDKBs in our assay, but this may reflect variable avidity of CDK antibodies and/or variable amounts of active CDK-cyclin complexes in cultured cells. Nevertheless, our results indicate that both CDKA and CDKB can phosphorylate Rep-MYBs in vitro.

Next, to identify the phosphorylation sites, we dissected the MYB3R3 coding region into three fragments, F1, F2 and F3 (Fig. 5b). CDKA immunoprecipitated from cultured cells phosphorylated F3, but neither F1 nor F2 (Fig. 5c). We then conducted liquid chromatography–tandem mass spectrometry (LC–MS/MS). Active CDKA-cyclin D3 complexes were produced in *E. coli*[35], and subjected to a kinase reaction with GST-MYB3R3. We identified six phosphorylated residues, among which Ser394 (S394), Thr407 (T407) and Ser461 (S461) were located in the F3 fragment (Supplementary Fig. 13). We also detected phosphorylation of the corresponding residues (S412, T427 and S493) in MYB3R5 (Supplementary Fig. 14). Note that these residues belong to the consensus motif of CDK phosphorylation sites (S/T-P). To identify in vivo phosphorylation sites, we tried to overexpress MYB3R3 in *Arabidopsis* plants and in protoplasts of *Arabidopsis* and tobacco cultured cells. However, we were unable to obtain sufficient protein for MS analysis, probably due to the highly unstable nature of phosphorylated MYB3R3.

In vitro kinase assays using His-MYB3R3 with alanine substitutions showed that phosphorylation level did not decrease for S394A or T407A, and not markedly for S461A (Fig. 5d), indicating the presence of multiple phosphorylation sites. A dramatic reduction in phosphorylation level was observed for S394A/T407A, S394A/S461A and S394A/T407A/S461A, but not for T407A/S461A (Fig. 5d). These results suggest that S394 is the major phosphorylation site, while T407 and S461 are also targeted by CDK. The persistent phosphorylation level observed for S394A indicates higher phosphorylation at T407 and/or S461 in this mutated protein (Fig. 5d), suggesting complementary roles of these three residues in being targeted by CDK. Impaired protein degradation was observed in vitro for His-MYB3R3 with

non-phosphorylatable alanine substitutions at all three sites (hereafter called MYB3R3AAA) (Fig. 4c), suggesting that CDK phosphorylation at these residues is indeed responsible for MYB3R3 degradation.

**Phosphomimic MYB3R3 cannot suppress root growth**. We then asked whether phosphorylation of Rep-MYBs is associated with the DNA damage response. As described above, GFP-fused MYB3R3 accumulates at a very low level in the root tip under non-stress conditions (Fig. 2c, d); however, *MYB3R3AAA-GFP* expressed under the *MYB3R3* promoter in *myb3r3* showed higher GFP fluorescence, and we observed less dramatic increases in the fluorescence levels following zeocin or MG132 treatment (Fig. 6a). This result was supported by immunoblotting using anti-GFP antibody (Fig. 6b), indicating that MYB3R3AAA-GFP accumulates highly even in the absence of DNA damage. We also introduced *ProMYB3R3::MYB3R3DDD-GFP*, which carries phosphomimic substitutions to aspartic acid at the three phosphorylation sites, into *myb3r3*. The level of MYB3R3DDD-GFP was very low in the absence of zeocin, and was not as highly elevated by zeocin treatment as that for *ProMYB3R3::MYB3R3-GFP* (Figs. 2c, d and 6a, b). This suggests that a decrease in phosphorylation of the three amino acids is necessary for DNA damage-induced protein accumulation of MYB3R3.

Kobayashi et al.[20] reported that loss of Rep-MYBs caused enhanced organ growth and ectopic cell division. In agreement with those results, the *myb3r3* mutant displayed slightly enhanced shoot growth (Fig. 6c). Whereas *myb3r3* expressing *MYB3R3DDD-GFP* showed a similarly enhanced phenotype, expression of *MYB3R3AAA-GFP* suppressed shoot growth as compared to WT (Fig. 6c). Slower growth was also noted in *myb3r3* roots harbouring *ProMYB3R3::MYB3R3AAA-GFP* (Fig. 6d). It is noteworthy that the average nuclear area of root meristematic cells expressing MYB3R3AAA-GFP was larger than of those expressing *ProMYB3R3::MYB3R3-GFP* (Fig. 3d). This suggests that a higher accumulation of MYB3R3AAA-GFP delayed cell cycle progression, thereby slowing down shoot and root growth.

In the presence of 2 μM zeocin, *myb3r3* expressing *MYB3R3-GFP* or *MYB3R3AAA-GFP* showed root growth inhibition to a similar extent to WT, while expression of *MYB3R3DDD-GFP* did not rescue the mutant phenotype, but rather led to tolerance to zeocin as observed in *myb3r3* (Fig. 6d, e). As described below, alanine substitutions at the three phosphorylation sites did not impair the ability of MYB3R3 to bind to target promoters or to repress target genes, suggesting that the three residues are not essential for protein function; rather, they control protein accumulation. Therefore, it is likely that the failure of MYB3R3DDD-GFP to rescue the mutant phenotype was due to a protein level that was too low to suppress G2/M-specific genes.

In *myb3r3* and *myb3r5*, the CDK inhibitor genes *SMR5* and *SMR7*, and the Act-MYB gene *MYB3R4*, were rapidly upregulated and downregulated by zeocin, respectively, as in WT (Supplementary Fig. 15). A previous analysis of global transcription and our qRT-PCR data for *MYB3R4* indicate that these transcriptional responses are dependent on SOG1[11] (Supplementary Fig. 16). This indicates that, in *myb3r3* and *myb3r5*, an initial reduction in CDK activity normally occurs by upregulation and downregulation of genes for CDK inhibitors and Act-MYB, respectively, in a SOG1-dependent manner, but it is not sufficient for growth arrest. This supports the above-mentioned idea that Rep-MYBs, which avoid CDK phosphorylation and accumulate highly under DNA stress, play an indispensable role in inhibiting cell division and organ growth.

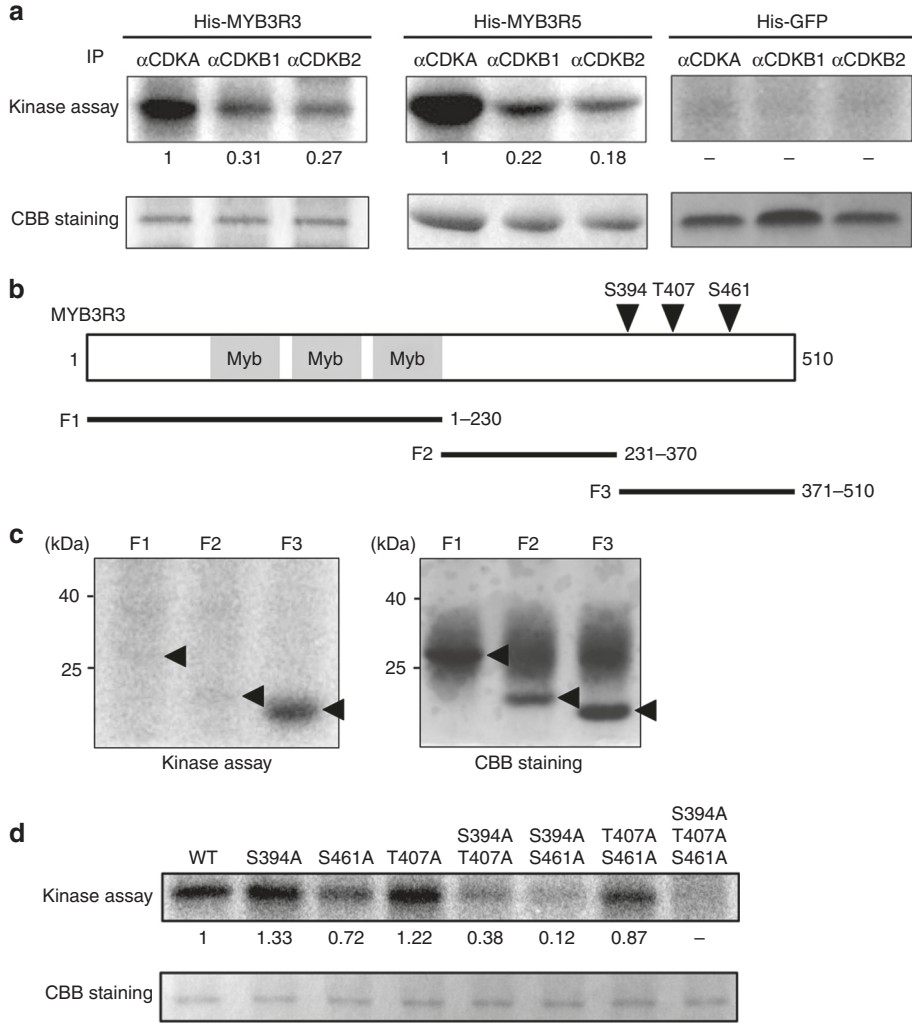

**Fig. 5** CDK phosphorylates MYB3R3 and MYB3R5. **a** In vitro kinase assay of His-MYB3R3 and His-MYB3R5. Protein extracts from *Arabidopsis* MM2d cultured cells were immunoprecipitated with specific antibodies against CDKA, CDKB1 and CDKB2, and assayed for kinase activity using His-MYB3R3 or His-MYB3R5 as substrate. His-GFP was used as a control. Relative intensity of each band is expressed as the fold change, normalized with respect to the band of substrates visualized by CBB staining. **b** Schematic representation of MYB3R3 harbouring three Myb domains. *Arrowheads* indicate CDK phosphorylation sites in the C-terminal region. *Black bars* indicate protein fragments used for kinase assay in **c**. **c** Kinase assay of partial fragments of MYB3R3. Protein extracts from MM2d cells were immunoprecipitated with the anti-CDKA antibody and assayed for kinase activity using His-tagged fragments of MYB3R3 (F1, F2 and F3) as substrate. *Arrowheads* indicate the position of each substrate. **d** Kinase assay of His-MYB3R3 with alanine substitutions. Protein extracts from MM2d cells were immunoprecipitated with the anti-CDKA antibody and assayed for kinase activity using His-MYB3R3 with alanine substitutions at CDK phosphorylation sites as substrate. Relative intensity of each band is expressed as the fold change, normalized with respect to the band of substrates visualized by CBB staining

**Rep-MYBs suppress G2/M-specific genes upon DNA damage.** We next examined whether DNA damage affects the binding of MYB3R3 to promoters of G2/M-specific genes. We focused on four genes identified as MYB3R3 targets: *KNOLLE*, *CYCB1;2*, *EDE1* and *IMK2*[20]. Chromatin immunoprecipitation (ChIP) using *ProMYB3R3::MYB3R3-GFP* seedlings showed that the promoter fragments of all four genes were highly enriched by anti-GFP antibody (Fig. 7a). The promoter of *CDKA;1*, whose expression is not specific to G2/M[31], did not show any enrichment, indicating the specificity of ChIP (Fig. 7a). When seedlings were treated with 2 μM zeocin for 24 h, MYB3R3 binding was significantly enhanced on the promoters of *KNOLLE* and *CYCB1;2*, but not *EDE1* or *IMK2*, relative to the non-treated control (Fig. 7b). We also tested other MYB3R3 target genes; MYB3R3 binding to the promoters of *PAKRP1*, *EHD2* and *PLE/MAP65-3* was highly elevated by zeocin treatment, while a slight or no increase was observed for those of *AUR2*, *PAKRP2*

and *CDC20.1* (Supplementary Fig. 17). These results indicate that MYB3R3, highly accumulated under DNA damage conditions, efficiently binds to the promoters of some of its target genes. The binding of MYB3R3 to the *KNOLLE* and *CYCB1;2* promoters showed a marked increase after 12 h and 18 h of zeocin treatment, respectively (Fig. 7c), which agrees with the observation that MYB3R3-GFP was abundant in the root tip after 12 h (Fig. 2a).

We next compared their binding to the *KNOLLE* and *CYCB1;2* promoters between MYB3R3-GFP, MYB3R3AAA-GFP and MYB3R3DDD-GFP. As shown in Fig. 8a, the binding of MYB3R3-GFP was elevated by zeocin treatment, while that of MYB3R3AAA-GFP was already high in the absence of zeocin. On the other hand, MYB3R3DDD-GFP did not display as dramatic an increase in promoter binding as that observed for MYB3R3-GFP even after 24 h of zeocin treatment. This result correlates with the levels of GFP fusion proteins accumulating in each line (Fig. 6b). Measurement of *KNOLLE* and *CYCB1;2* transcripts

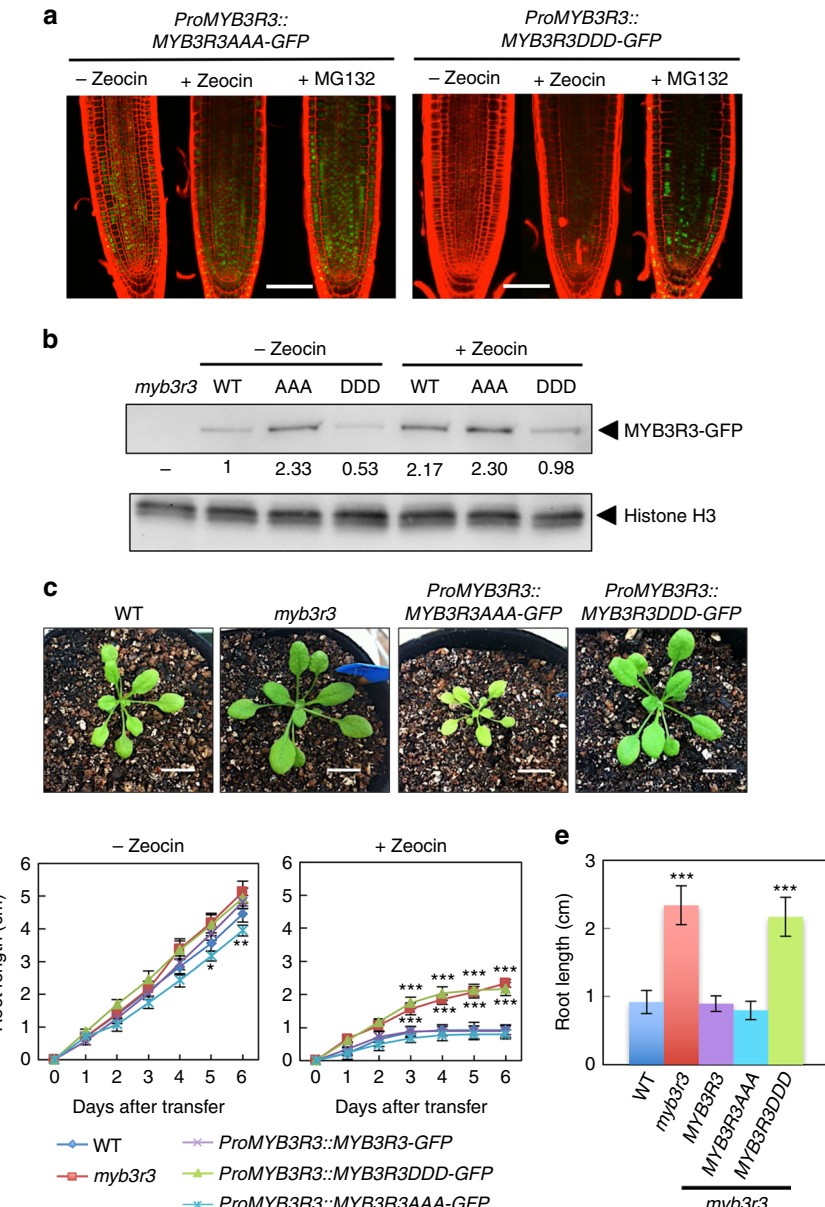

**Fig. 6** Phosphorylation state of MYB3R3 is associated with protein accumulation and zeocin response. **a** Confocal microscopy images of *myb3r3* root tips harbouring *ProMYB3R3::MYB3R3AAA-GFP* or *ProMYB3R3::MYB3R3DDD-GFP*. Five-day-old seedlings were treated with or without 2 μM zeocin, or with 50 μM MG132, for 24 h. *Scale bars*, 100 μm. **b** Protein level of MYB3R3-GFP in roots. Ten-day-old *myb3r3* seedlings harbouring *ProMYB3R3::MYB3R3-GFP* (WT), *ProMYB3R3::MYB3R3AAA-GFP* (AAA) or *ProMYB3R3::MYB3R3DDD-GFP* (DDD) were treated with or without 2 μM zeocin for 24 h, and 40 μg of total protein extracted from roots was immunoblotted with anti-GFP or anti-histone H3 antibodies. Protein extract from *myb3r3* was used as a control. Relative levels of MYB3R3-GFP are expressed as the fold change, normalized with respect to the band of histone H3. **c** Eighteen-day-old seedlings of WT, *myb3r3* and *myb3r3* carrying *ProMYB3R3::MYB3R3AAA-GFP* or *ProMYB3R3::MYB3R3DDD-GFP*. *Scale bars*, 2 cm. **d, e** Root growth of WT, *myb3r3* and *myb3r3* carrying *ProMYB3R3::MYB3R3-GFP*, *ProMYB3R3::MYB3R3AAA-GFP* or *ProMYB3R3::MYB3R3DDD-GFP*. Five-day-old seedlings were transferred to medium with or without 2 μM zeocin, and root length was measured for 6 days **d**. Root length after 6 days of zeocin treatment is shown in **e**. Data are presented as mean ± SD (n > 30). Significant differences from WT were determined by Student's *t*-test: *$P < 0.05$; **$P < 0.01$; ***$P < 0.001$

showed that they decreased after 24 h of zeocin treatment in *myb3r3* expressing *MYB3R3-GFP* or *MYB3R3AAA-GFP* to levels comparable to those in WT plants (about 60 and 80% reduction for *KNOLLE* and *CYCB1;2*, respectively). However, in *myb3r3* expressing *MYB3R3DDD-GFP*, the transcript levels were less drastically decreased (about 40–50% reduction for both genes) (Fig. 8b). These results suggest that the protein level of MYB3R3 determines its binding to target promoters and the extent of repression of target genes.

When WT seedlings were treated with 2 μM zeocin, transcripts of *KNOLLE* and *CYCB1;2* were reduced and reached a stable level after 18 h (Fig. 9). Those of *EDE1* and *IMK2* also decreased but reached the stable level after 12 h. In *myb3r3* and *myb3r5* mutants, transcript accumulation of *EDE1* and *IMK2* was almost the same as that in WT, but mRNA levels of *KNOLLE* and *CYCB1;2* did not further decrease beyond 12 h of zeocin treatment (Fig. 9). A similar tendency was also observed in *myb3r3* for *PAKRP1*, *EHD2* and *PLE/MAP65-3*, but not for *AUR2*, *PAKRP2* or *CDC20.1* (Supplementary Fig. 18). These

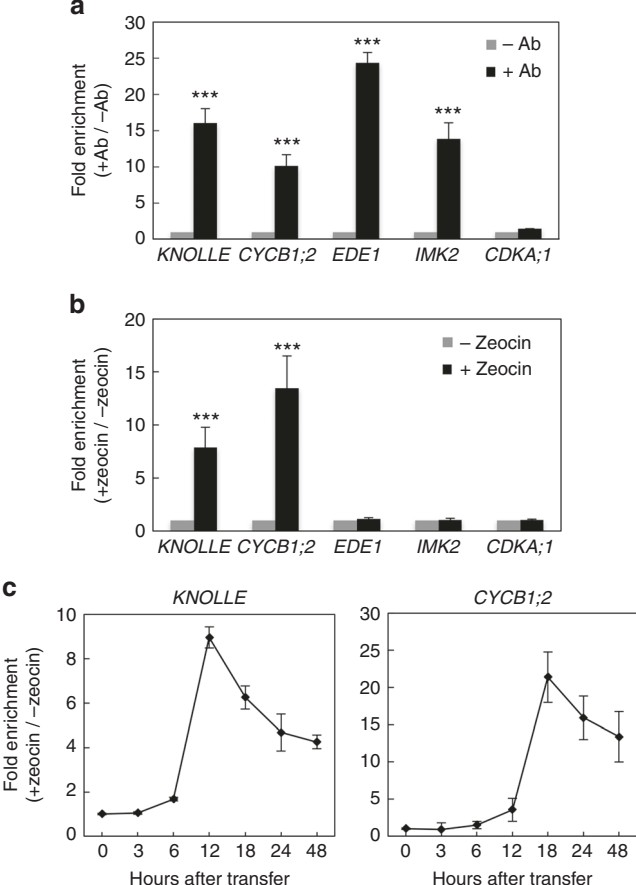

**Fig. 7** DNA damage promotes the binding of MYB3R3 to *KNOLLE* and *CYCB1;2* promoters. **a** ChIP-PCR analysis of G2/M-specific genes. Chromatin bound to MYB3R3 was collected by immunoprecipitation with (+Ab) or without (−Ab) anti-GFP antibodies using roots of 10-day-old *myb3r3* plants carrying *ProMYB3R3::MYB3R3-GFP*. Fold enrichment of each promoter region was determined by normalizing the recovery rate against that of samples immunoprecipitated without the antibody. *CDKA;1* was used as a control that is expressed throughout the cell cycle. Data are presented as mean ± SD of three biological replicates. Significant differences from the control immunoprecipitated without the antibody were determined by Student's *t*-test: ***$P < 0.001$. **b**, **c** ChIP-PCR analysis using zeocin-treated roots. Ten-day-old *myb3r3* seedlings carrying *ProMYB3R3::MYB3R3-GFP* were treated with or without 2 μM zeocin for 24 h (**b**) or for the indicated times (**c**), and used for ChIP assay. Fold enrichment of each promoter region was determined by normalizing the recovery rate against that of samples without zeocin treatment. Data are presented as mean ± SD of three biological replicates. Significant differences from the control without zeocin treatment were determined by Student's *t*-test: ***$P < 0.001$ (**b**)

results suggest that Rep-MYBs function in further suppressing a particular set of G2/M-specific genes at a later stage of the DNA damage response (beyond 12 h of zeocin treatment in our experimental system), thus blocking cell cycle progression.

## Discussion

Previous reports demonstrated that tobacco and *Arabidopsis* Act-MYBs are phosphorylated and activated by CDK[18, 29]. Since mitotic cyclins, one of the targets of MYB3Rs, form active CDK-cyclin complexes, Act-MYBs are further phosphorylated and activated through a positive feedback loop, thus causing a burst of G2/M-specific gene expression (Fig. 10, *left*). In this study, we found that Rep-MYBs are targeted for degradation by

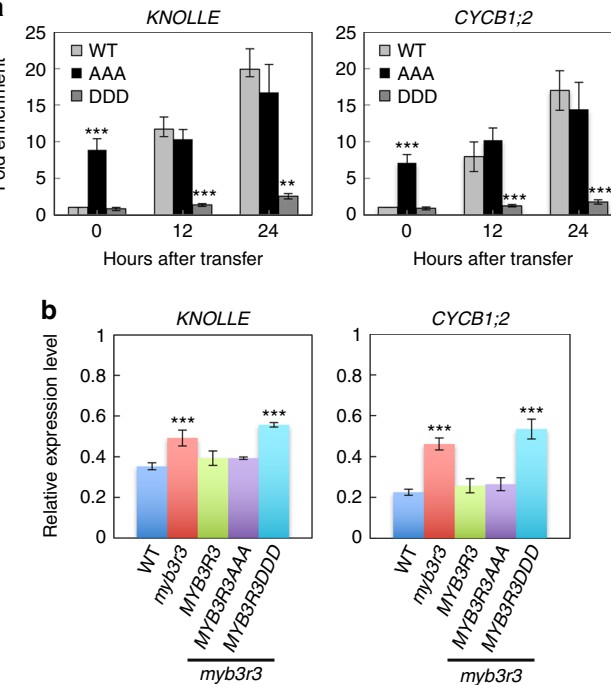

**Fig. 8** Non-phosphorylated form of MYB3R3 represses the *KNOLLE* and *CYCB1;2* promoters. **a** ChIP-PCR analysis of *KNOLLE* and *CYCB1;2*. Ten-day-old seedlings of *myb3r3* carrying *ProMYB3R3::MYB3R3-GFP* (WT), *ProMYB3R3::MYB3R3AAA-GFP* (AAA) or *ProMYB3R3::MYB3R3DDD-GFP* (DDD) were treated with 2 μM zeocin for the indicated times, and used for ChIP assay. Chromatin bound to MYB3R3 was collected by immunoprecipitation with anti-GFP antibodies. Fold enrichment of each promoter region was determined by normalizing the recovery rate against that of *ProMYB3R3::MYB3R3-GFP* without zeocin treatment. Data are presented as mean ± SD of three biological replicates. Significant differences from *ProMYB3R3::MYB3R3-GFP* were determined by Student's *t*-test: **$P < 0.01$; ***$P < 0.001$. **b** Quantitative RT-PCR analysis of *KNOLLE* and *CYCB1;2*. Five-day-old seedlings of WT, *myb3r3* and *myb3r3* carrying *ProMYB3R3::MYB3R3-GFP*, *ProMYB3R3::MYB3R3AAA-GFP* or *ProMYB3R3::MYB3R3DDD-GFP* were transferred to medium with 2 μM zeocin and grown for 24 h. Total RNA was isolated from roots and subjected to quantitative RT-PCR analysis. The expression levels were normalized to that of *ACTIN2*, and are indicated as relative values, with that for the time of transfer to zeocin-containing medium set to 1. Data are presented as mean ± SD of three biological replicates. Significant differences from the WT control were determined by Student's *t*-test: ***$P < 0.001$

proteasomes. Treatment with the CDK-specific inhibitor roscovitine enhanced accumulation of Rep-MYBs, indicating that CDK activity is required for Rep-MYB degradation. Indeed, Rep-MYBs were phosphorylated by CDKs in vitro, and phospho-mimic or alanine substitutions at the phosphorylation sites led to lower or higher accumulation of MYB3R3 in roots, respectively. These data suggest that, in the absence of DNA stress, CDK phosphorylates Rep-MYBs and promotes proteasomal degradation (Fig. 10, *left*). Kobayashi et al.[20] reported that Rep-MYBs repress transcription along the cell cycle except for G2/M, and our measurement of nuclear area in the meristem showed that MYB3R3 protein preferentially accumulates around early S phase. Since CDK activity is high at G2/M phase[36], it is probable that CDK-mediated control of protein degradation is involved in oscillation of Rep-MYB accumulation during the cell cycle.

In response to DNA damage, the ATM/ATR-SOG1 pathway is activated, and SOG1 induces the expression of CDK suppressors such as SMRs, thereby reducing CDK activity[14, 37]. We previously

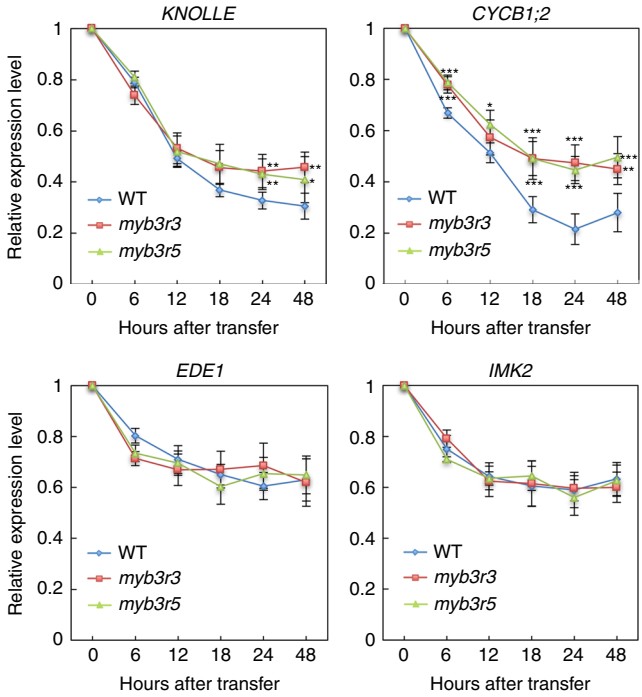

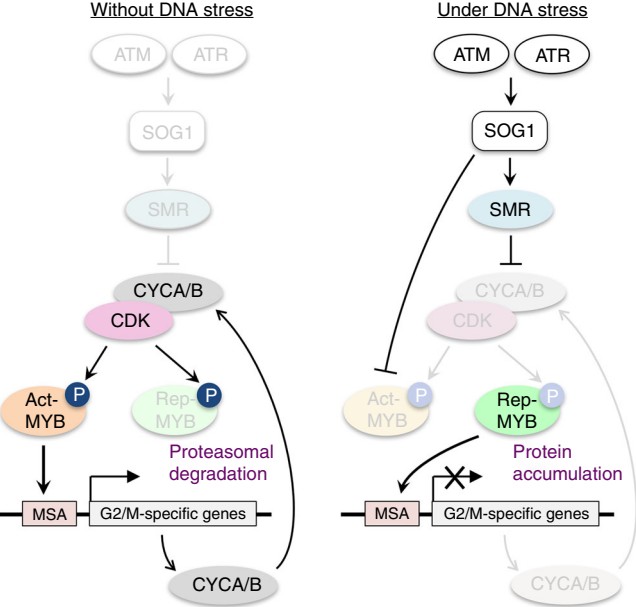

**Fig. 9** Quantitative RT-PCR analysis of G2/M-specific genes. Six-day-old seedlings were transferred to medium containing 2 μM zeocin and grown for the indicated times. Total RNA was isolated from the region 1 cm from the root tip. The expression levels were normalized to that of *ACTIN2*, and are indicated as relative values, with that for the time of transfer to zeocin-containing medium set to 1. Data are presented as mean ± SD of three biological replicates. Significant differences from WT were determined by Student's *t*-test: *$P < 0.05$; **$P < 0.01$; ***$P < 0.001$

**Fig. 10** Model for Rep-MYB-mediated repression of G2/M-specific genes under DNA stress conditions. Among the G2/M-specific genes that are controlled by MYB3Rs are mitotic cyclins, such as CYCA and CYCB (CYCA/B). CDK-CYCA/B generates a positive feedback loop with Act-MYBs, which enhance the expression of G2/M-specific genes. Under DNA stress conditions, the ATM/ATR-SOG1 pathway disrupts this feedback loop by up- and down-regulating CDK suppressors (i.e., SMRs) and Act-MYBs, respectively. As a result, Rep-MYBs are liberated from CDK phosphorylation and proteasomal degradation, and function in repressing a particular set of G2/M-specific genes

reported that *MYB3R4* expression is downregulated by DSBs[14], and here we revealed that SOG1 is required for a rapid decrease of *MYB3R4* transcripts. As a result, Act-MYB-mediated transcription of G2/M-specific genes is suppressed, and lowered CDK activity liberates Rep-MYBs from proteasomal degradation (Fig. 10, *right*). In support of this idea, we observed zeocin-induced accumulation of Rep-MYBs, which was cancelled by introducing phosphomimic substitutions at the phosphorylation sites. Rep-MYBs then bind to the MSA element and repress the expression of G2/M-specific genes (Fig. 10, *right*). We showed that this repression occurs at a later stage of the DNA damage response and is essential for inhibiting cell division. We also found that the *myb3r3* and *myb3r5* mutants exhibited less zeocin-induced stem cell death, which is also a programmed response involving the ATM/ATR-SOG1 pathway[15, 16]. Therefore, stem cell death is most likely another outcome of Rep-MYB-mediated cell cycle arrest.

Our results indicate that Rep-MYBs function in repressing a particular set of G2/M-specific genes under DNA stress. There is no apparent difference in the core MSA element between target and non-target genes, and differential binding activity of Rep-MYBs may therefore be dependent on promoter contexts of individual G2/M-specific genes. Kobayashi et al.[20] demonstrated that Rep-MYBs form a complex with retinoblastoma-related protein and the E2F transcription factor E2FC, indicating that plants have the multiprotein complex known as DREAM/dREAM-like complex in vertebrates and *Drosophila*. In mammalian cells, the transition from quiescence to mitotic state is accompanied by the release of some subunits of the DREAM complex, in conjunction with the recruitment of the Myb transcription factor B-MYB, which activates transcription of various

genes that are essential for mitosis[38]. It is therefore probable that *Arabidopsis* Rep-MYBs belong to a distinct DREAM complex under DNA stress conditions, and that another subunit participates in controlling target specificity together with Rep-MYBs. Biochemical and genetic studies on the DREAM complex may elucidate the mechanism regulating a distinct set of G2/M-specific genes under DNA damage conditions.

During protein degradation by the 26S proteasome, phosphorylation of target proteins provides a recognition signal for an E3 ubiquitin ligase, especially for SCF (SKP1-CUL1-F-box) complexes[39]. In the case of Sic1, one of the CDK inhibitors in budding yeast, phosphorylation of a minimum of six residues by Cdc28 sets a threshold for SCF recognition, thereby providing a sensitive switch mechanism for the induction of the G1/S transition[40]. Here, we show that CDKs can phosphorylate Rep-MYBs in vitro, and that CDK activities are required for their degradation by proteasomes. On the other hand, CDK is known to phosphorylate and activate Act-MYBs[18, 29]. This implies that CDK acts on Act- and Rep-MYBs in opposite directions, and that once CDK activities decrease in response to DNA damage, the mitotic state can switch to the quiescent state in a rapid and precise manner. It is therefore possible that other stresses that cause a reduction in CDK activity also induce MYB3R-mediated inhibition of cell division. Further studies will identify the E3 ubiquitin ligase that recognizes Rep-MYBs and reveal how it is involved in stress responses by controlling the cell cycle.

## Methods
**Plant materials and growth conditions**. WT and mutant plants used in this study are all in the Col-0 background. *lig4*, *sog1-1*, the T-DNA insertion mutants *myb3r3-1* (SALK_041111), *myb3r3-2* (GABI 546A07), *myb3r4-2* (SALK_034806) and *myb3r5-1* (SALK_031972), and the transgenic line carrying *ProMYB3R3::MYB3R3-*

GFP were described previously[11, 18, 20, 24]. *myb3r4-3* (GABI 634B01) and *myb3r5-2* (GABI 252G10) were also used in this study. To generate *ProMYB3R5::MYB3R5-GFP*, the promoter and the coding region of *MYB3R5* were PCR-amplified from genomic DNA using primers listed in Supplementary Table 1, and cloned into the Gateway entry vector pDONR221 (Invitrogen) by a BP reaction. An LR reaction was performed with the destination vector pGWB550[41] to generate a binary vector carrying the fusion construct with *GFP*. Alanine or aspartic acid substitutions were introduced by PCR using primers listed in Supplementary Table 1. Plants were grown in MS medium under continuous light conditions at 22 °C. Chemicals used in this study are zeocin (Invitrogen), bleomycin (Wako), bleocin (Wako), hydroxyurea (Nacalai Tesque), aphidicolin (Sigma), boric acid (Sigma), MG132 (Biomol) and roscovitine (Wako). Gamma-ray irradiation was conducted with a $^{137}$Cs source (Radiation Biology Center, Kyoto University). For counting the number of true leaves, seeds were grown in MS medium for 10 days.

**Microscopy**. Seedlings were immersed in 0.1 mg ml$^{-1}$ propidium iodide solution for 3 min, and observed with a confocal laser scanning microscope (FV1000, Olympus). Root meristem size was measured by counting the number of cortex cells between the quiescent centre and the first elongated cell. GFP fluorescence was quantified by AxioVision Rel. 4.8 software (Carl Zeiss).

**Nuclear area measurement**. Roots were fixed in MTSB (5 mM PIPES, 0.5 mM EGTA, 0.35 mM MgSO$_4$ 7H$_2$O, 8.9 mM KOH, pH 8.0) containing 3% (w/v) paraformaldehyde and 0.5% (v/v) Triton X-100 for 20 min at room temperature. After washing with phosphate-buffered saline (PBS), cell walls were partially digested in an enzyme solution containing 0.05% (w/v) pectolyase (Kyowa Kasei) and 0.15% (v/v) Triton X-100 for 2 h at 30 °C. The roots were then washed with PBS and stained with 1 µg ml$^{-1}$ 4′,6-diamidino-2-phenylindole (DAPI) for 15 min at room temperature. Fluorescence images of epidermal trichoblast cells in the root meristem were taken from a Z-stack of 20 confocal images at 0.4 µm intervals, and the resulting images were processed using ImageJ software to measure the nuclear area.

**Double staining with EdU and DAPI**. EdU staining was conducted with a Click-iT Plus EdU Alexa Fluor 647 Imaging Kit (Thermo Fisher Scientific) according to the manufacturer's instruction. In brief, seedlings were immersed in liquid MS medium containing 10 µM EdU for 60 min, and fixed in MTSB containing 3% (w/v) paraformaldehyde and 0.5% Triton X-100 for 20 min. After washing twice with PBS, samples were incubated in the Click-iT reaction cocktail for 30 min in the dark. The reaction cocktail was then removed, and samples were washed with PBS, followed by cell wall digestion and DAPI staining as described for nuclear area measurement.

**Pulse labelling with EdU**. Four-day-old WT seedlings were grown on solid medium containing different concentrations of zeocin for 12 h and transferred to liquid MS medium containing zeocin and 20 µM EdU, followed by a 15-min incubation. After washing with MS medium, seedlings were again grown on zeocin-containing medium for different chase time periods, and subjected to double staining with EdU and DAPI as described above.

**Immunoblotting**. Total protein extracted from root tips was separated by 10% sodium dodecyl sulphate-polyacrylamide gel electrophoresis (SDS-PAGE) and blotted onto a polyvinylidene difluoride membrane (Millipore). Immunoblotting was conducted with anti-GFP antibody (11814460001, Roche), anti-α-tubulin antibody (ab4074, Abcam) or anti-histone H3 antibody (AS10 710, Agrisera) at a dilution of 1:2000, 1:3000 or 1:5000, respectively. Clarity Western ECL Substrate (Bio-Rad Laboratories) was used for detection. The intensity of each band was measured by Image Lab software (Bio-Rad Laboratories), and relative values were calculated by normalizing with respect to the bands of controls. Uncut blot images are shown in Supplementary Figs. 19, 20 and 22.

**In vitro kinase assay**. Whole or partial *MYB3R* open reading frames were cloned into the pENTR/D-TOPO vector (Invitrogen), and then transferred to the pDEST17 vector (Invitrogen) using LR Clonase (Invitrogen) to be in-frame with the 6× His tag. Alanine substitutions were introduced by PCR using primers listed in Supplementary Table 1. His-tagged MYB3R proteins were purified with Ni Sepharose 6 Fast Flow (GE Healthcare) according to the manufacturer's instruction. For in vitro kinase assay, CDKs were immunoprecipitated from MM2d cultured cells with specific antibodies, and the immunoprecipitates were incubated with 1 µg His-tagged substrates in 10 µl assay buffer (50 mM Tris-HCl, 15 mM MgCl$_2$, 5 mM EGTA, 1 mM dithiothreitol, 0.01 mM ATP, 0.185 MBq of [γ-$^{32}$P] ATP, pH 7.5) at 30 °C for 15 min. The reaction was stopped by the addition of sample buffer for SDS-PAGE, boiled for 5 min, and loaded onto a polyacrylamide gel. Phosphorylated proteins were detected with an imaging analyzer (BAS-1800, Fujifilm). Uncut autoradiography images are shown in Supplementary Fig. 21.

**In vitro degradation assay**. One hundred micrograms of root extracts from *Arabidopsis* seedlings was incubated with 50 ng of His-MYB3R3 or His-

MYB3R3AAA in a buffer containing 25 mM Tris-HCl (pH 7.5), 10 mM MgCl$_2$, 5 mM dithiothreitol, 10 mM NaCl and 10 mM ATP. After protein separation by SDS-PAGE, immunoblotting was conducted using anti-His antibodies (11922416001, Roche) at a 1:5000 dilution and an ECL Western Blotting Detection Kit (GE Healthcare).

**Chromatin immunoprecipitation**. Two grams of 10-day-old *ProMYB3R3:: MYB3R3-GFP* seedlings were collected and fixed in 100 ml of 1% (w/w) formaldehyde under vacuum for 30 min. After washing twice with ice-cold 0.2 M glycine, the plant materials were ground into powder with liquid nitrogen and lysed in 40 ml nuclear isolation buffer [1 M hexylene glycol, 50 mM Tris-HCl (pH 7.5), 10 mM MgCl$_2$, 0.5% Triton X-100, 5 mM 2-mercaptoethanol, 1× protease inhibitor cocktail (Roche)]. Nuclei were separated by filtration through Miracloth (Calbiochem), and isolated nuclei were suspended in 6 ml lysis buffer (50 mM Tris-HCl (pH 7.5), 100 mM NaCl, 0.1% Triton X-100, 1× protease inhibitor cocktail (Roche)) and sonicated. Chromatin complexes bound to MYB3R3-GFP were immunoprecipitated with anti-GFP antibody (GF200, Nacali Tesque) at a 1:1000 dilution. To quantify the precipitated chromatin, gene-specific primers (Supplementary Table 1) were used for real-time qPCR.

**Quantitative RT-PCR**. Total RNA was extracted from *Arabidopsis* roots with an RNeasy Plant Mini Kit (QIAGEN). First-strand cDNAs were prepared from total RNA using the Superscript II First-Strand Synthesis System (Invitrogen) according to the manufacturer's instruction. Quantitative PCR was performed with a THUNDERBIRD SYBR qPCR Mix (Toyobo) with 100 nM primers and 0.1 µg of first-strand cDNAs. Primer sequences are listed in Supplementary Table 1. PCR reactions were conducted with the LightCycler 480 Real-Time PCR System (Roche) under the following conditions: 95 °C for 5 min; 45 cycles of 95 °C for 10 s, 60 °C for 10 s and 72 °C for 15 s. *ACTIN2* (At3g18780) was used as a control.

**Identification of phosphorylation sites**. For production of CDKA;1-CYCD3;1 complexes, *E. coli*. BL21 strain was transformed with the two plasmids, pCDFDuet-CDKA;1 and pHGGWA-CYCD3;1, which carry *StrepIII-CDKA;1* and *GST-CYCD3;1*, respectively[35]. Cells were grown in 50 ml of LB medium containing 50 µg ml$^{-1}$ ampicillin and 50 µg ml$^{-1}$ spectinomycin at 37 °C to OD$_{600}$ of 0.6. The production of proteins was induced by adding IPTG to 0.3 mM, and cells were cultured overnight at 18 °C. After centrifugation, the cell pellet was resuspended in 2.5 ml binding buffer (50 mM NaH$_2$PO$_4$, 100 mM NaCl, 10% (v/v) glycerol, 25 mM imidazole, pH 8.0) and lysed by sonication. The cell slurry was then centrifuged at 10,000×*g* for 40 min at 4 °C, and the supernatant was applied onto an Econopack column (Bio-rad) packed with 300 µl Ni-NTA resin (Qiagen). After washing sequentially with 3 ml binding buffer, CDKA;1–CYCD3;1 complexes were eluted with 600 µl elution buffer (50 mM NaH$_2$PO$_4$, 100 mM NaCl, 10% (v/v) glycerol, 225 mM imidazole, pH 8.0). For phosphorylation reaction, the buffer was exchanged to the kinase buffer (50 mM Tris-HCl, 10 mM MgCl$_2$, 1 mM EGTA, pH 7.5) containing 1× protease inhibitors cocktail (Roche). *MYB3R3* and *MYB3R5* open reading frames were transferred from the pENTR/D-TOPO vector to the pDEST15 vector (Invitrogen) using LR Clonase to be in-frame with the GST tag. GST-MYB3R3 and GST-MYB3R5 were purified with Glutathione Sepharose 4B (GE Healthcare) according to the manufacturer's instruction.

A phosphorylation reaction was performed using 1 mg of GST-MYB3R3 and GST-MYB3R5 as substrate. After separation of reaction mixtures by SDS-PAGE, proteins were stained with Coomassie Brilliant Blue, and the bands corresponding to GST-MYB3R3 and GST-MYB3R5 were subjected to in-gel digestion using Sequencing Grade Modified Trypsin (Promega). LC–MS/MS analysis was performed using an LTQ-orbitrap XL-HTC-PAL system (Thermo Fisher Scientific)[42]. Obtained spectra were compared with a protein database (TAIR10) using the Mascot server (version 2.4). The Mascot search parameters were as follows: ion-score cut-off with the threshold at 0.05; peptide tolerance at 10 p.p.m.; MS/MS tolerance at ± 0.8 Da; peptide charge of 2$^+$ or 3$^+$; trypsin as enzyme allowing up to two missed cleavages; carbamidomethylation on cysteine as a fixed modification and oxidation on methionine and phosphorylation on serine, threonine and tyrosine as variable modifications.

**Data availability**. All data supporting the findings of this study are available within the article and its Supplementary Information files or from the corresponding author on request.

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

## Acknowledgements

We would like to thank Anne B. Britt for sog1-1 seeds. This work was supported by MEXT KAKENHI (Grant numbers 22119009 and 24657033), JSPS KAKENHI (Grant numbers 26291061 and 26650099) and JST, CREST (Grant number JPMJCR12B2).

## Author contributions

P.C., H.T., M.I. and M.U. designed the research. P.C., N.T., H.T., R.K., Y.F. and K.K. performed the experiments and analyzed data. P.C., N.T., H.T., Y.F. and M.U. wrote the article.

## Additional information

**Competing interests:** The authors declare no competing financial interests.

