## [Peer Review File · Nature Communications]

Reviewers' comments:

Reviewer #1 (Remarks to the Author):

Chen et al. present a nicely prepared manuscript describing the role of MYB transcription factors in response to DNA damage in Arabidopsis. Overall, the paper is sufficiently clear and the experiments are mostly well done. The impact of the paper in the field will be strong.

My main concern with the paper is perhaps a more direct demonstration that the factors described directly affect the G2/M transition. The experiments use "knockout" mutants to demonstrate resistance (or hypersensitivity in the case of r4) to DNA damage. The authors use a range of different damaging agents, but I would have liked to see inclusion of an agent such as camptothecin that more specifically induces DSBs during S-phase. This might be more informative than a general DSB inducing agent or replication inhibitor. More importantly, the authors provide nice data regarding induction of G2-specific genes, but in my mind this is indirect evidence for G2 arrest in dividing cells. The authors are careful in their model to state induction of G2/M specific genes, but in the discussion say this leads to G2 arrest. This may likely be the case, but I don't think the experiments directly demonstrate this. Are G2/M genes being turned on but would we see S-phase arrest? Perhaps the authors could show the cellular DNA content using cell sorting of GFP+ cells shortly after DNA damage in their MYB3R3-GFP or MYB3R3AAA-GFP lines? If their assumption is correct, we would expect these cells to be mostly arrested as 4N. If the authors believe this type of experiment is not possible, then they should at least make it more clear in the discussion that direct demonstration of the G2/M arrest still needs to be explored. Or perhaps the authors could provide evidence in the literature to make a stronger case for their assumption.

Minor comments:

Line numbers would be welcome in the manuscript. Also inclusion of figure legends with the figures is helpful.

Do the myb3/5 mutants show the gamma-plantlet phenotype described for the original sog mutant?

Page 3, line ??, last sentence in first paragraph, "(refs 12, 13)" should be standard formatting?

Page 3, second paragraph, "that of" can be eliminated, second to last sentence.

Page 14, second paragraph, "showed" change to "show".

Quantification of GFP cells in Figure 2/4 would be helpful to support your arguments of differences in expression. We use confocal microscopy and simply quantify GFP signals using standard software.

Some of the graphs use symbols in them, but not in the legends. This made it difficult to distinguish the various lines in black and white prints.

Reviewer #2 (Remarks to the Author):

This manuscript presents a very interesting study providing important new understanding of the cell-cycle response to DNA damage in plants. The authors use a variety of approaches to generate a significant amount of data supporting their model of the Rep-MYB transcription factors in the link between the DNA damage response and cell-cycle regulation.

This is a very good and coherent manuscript presenting convincing data and a solid interpretation.

I have one general query concerning the choice to place a lot of the figures in the supplemental data. I assume this is because of space considerations, but the editor might consider moving some of the Suppl. Figs to the main text. This is not essential, but I feel that the model (Suppl Fig 12) really should be in the article itself.

I do have a some specific comments:

1. page 2, line 1 - This first sentence needs rephrasing. Perhaps fuse it with the next sentence to give "Inhibition of cell division is an active response to DNA damage..."
2. page 2, line 16 - It is of course true that environmental influences cause DNA damage, but cellular processes (replication, ROS, ...) also do so. Only mentioning environmental effects is a pity.
3. page 4, line 17 -the authors should specify that their work on DSBs and G2 arrest is about Arabidopsis.
4. page 4, line 19 - These mutants are properly referenced in the Methods section and have been previously published, but I feel that a short description (more than the word "knockout") would be helpful to readers here.
5. page 5, line 6. suppl fig 1a - the myb3r3 and sog1 myb3r3 mutants reach a plateau at 5-6 days. This does not seem to be the case for the sog1 plants? I'm not sure that saying that they are "comparable" is enough here.
6. Page 5, Line 9, suppl 1b : a similar effect is visible in the myb3r3 myb3r5 double mutant, but not in the singles (not in myb3r3 in this experiment?). The authors should provide some discussion of this.
7. Page 5, Lines 9-10. The similar phenotypes of the two single and the double mutants shows that both (not either) of them are required and imply epistasis of the two proteins (same pathway).
8. Page 5, Lines 10-12 - this sentence is very complicated. Perhaps replace it with: "Observation of the root tip showed that increasing zeocin concentrations resulted in stronger (myb3r4) or weaker (myb3r3, myb3r5) reduction of the meristem size compared to the WT (Fig. 1c; Supplementary Fig. 2)."
9. Page 5, Line 20 - I suggest replacing "...displayed tolerance..." with "...were significantly more tolerant..."

Reviewer #3 (Remarks to the Author):

This manuscript by Chen et al provides insight in Rep-MYB3R proteins and cell cycle arrest under DNA damage, linking this with CDKA and phosphorylation of the Rep-MYB3Rs. While overall the genetic and biochemical data presented are solid (and the manuscript is largely clearly written), they are incomplete and premature to draw final conclusions. I have the following comments/suggestions:

- CDK targets both Act and Rep-MYB3Rs. How is this different? Both CDKA and CDKB? What is the effect on stability with respect to MYB3R4?
- In this context, a more complete analysis on R3, R4 and R5 for the majority of the experiments is required (regarding the Rep and Act behavior), e.g. expression pattern, binding to targets, double mutants, etc
- To further connect CDK activity to MYB3R activity, how does, for example, roscovitine impact target gene expression in absence/presence of zeocin?
- How is the whole cascade connected to CDKA/B on a genetic level? Phenotype of cdka/b mutants, the MYB3R phosphostatus and the impact on the target gene expression in the absence presence of A or D variants (complementation)?
- I do not fully follow the logic of the A substitution data. To me it seems that S461A has decreased

activity. In addition, it seems that S394A is the dominant target of CDKA (based on the double and triple combinations, only activity is lost when S394A is present), but as a single variant there is still activity. This needs to be carefully looked into and better explained. In this context, studying full AAA variants in the context of stability and activity might lead to some misinterpretation, and also the effect of single A and D variants on protein degradation, target gene expression, zeocin impact on stability, etc needs to be investigated. ChIP-PCR should be performed on AAA and DDD variants (also in absence/presence of zeocin).

- To support the model of competitive activity of (phosphorylated) Rep and Act-MYB3Rs, using ChIP-PCR in respective mutant backgrounds, but also using MYB3R A and D variants (for the importance of phosphorylation) is necessary.
- In the discussion it is not clear what has been shown previously (references), what is new here and what is speculation. This section needs to be thoroughly rewritten and focus on those aspects of the model where actual data is available for. The interaction of Rep-MYBs with RBR and E2F was mentioned, but does this depend on the phosphostatus of the Rep-MYBs?

Minor comments

- With respect to the data in Fig 2c and 4a, a comparative quantification (in one experiment) is needed for WT, AAA and DDD.
- In the figure legends (and/or methods) it should be indicated what $n = 3$ stands for in the context of PCR (biological or technical repeats). One should at least use 3 biological repeats.
- Regarding the figures (graphs), it might be useful to indicate the % reduction with respect to a phenotype such as root length. In addition, the labels need to be more clearly indicated in color and shape for wt and mutants in the legend. Given the overlap between the various lines in Figures (such as 4c), it might be useful to also show a bar diagram of 1 time point (with maximal difference).
- For various blots, the ratio relative to the control should be calculated. Also the loading control on Figure S5 needs to be shown.

Reviewer #4 (Remarks to the Author):

The work from Chen and co-workers focusses on three MYB3R proteins (MYB3R3, R4, and R5) and their role in plant development and cell cycle regulation. The authors can show that sensitivities towards DSB-inducing toxins are changed in null mutants that lack these transcription factors. They can further show that R3 and R5 are phosphorylated through three different CDKs purified from plant extracts. R3 and R5 are instable in a proteasome- (based on MG132 treatment), DSB- (Zeocin-treatment), and CDK dependent manner (roscovitine treatments) in vitro, and that mutation of three serine residues that are phosphorylated in R3 to alanine stabilizes the corresponding protein in vitro. Overall the work from Chen and co-workers is very interesting and provides new insights on the role and regulation of MYB3R transcription factors in plants and in context with genotoxic stress.

Comments.

Reading through the manuscript I find that the authors relate very often to cell cycle, however, most evidence (except for the very last part) is only provided through cell number, and this also only in the very first figure (1c). Personally I think that this should have been done in more elaborated way/more thoroughly to support statements, especially since the authors strongly correlate their findings with cell cycle. Root elongation as it was most often done by the authors is to my opinion not a very good indicator for cell cycle problems.

Can the authors explain why they only investigated three out of five myb3r null mutants, and also only a single allele for each of these in this work? I assume that all of these mutants have been

backcrossed at least twice? Maybe the authors can add these statements in the method and result part where suitable

On P. 4: I would disagree with the statements 'Zeocin tolerance was comparable between myb3r3, sog1-1 and the myb3r3 sog1-1' based on what is show in Supp. Fig. 1a, day 6. myb3R3 looks more sensitive compared to sog1-1 or the double. Likewise, I would disagree with the statement 'The myb3r3 myb3r5 double mutant was tolerant to zeocin to an extent similar to each single mutant', the graph in Supp Fig. 1b at day 6 does not support this statement. Furthermore, why did the authors only generate a sog1-1myb3r3 double? Why has this not done for the others? How do these mutants behave in a dose-dependent experiments trying different concentrations of Zeocin?

On P. 5 the authors state 'Observation of the root tip showed that increasing zeocin concentration resulted in stronger or weaker reduction of the meristem size in myb3r4 or myb3r3'. Can the authors refer here to cell number rather than size, as the latter was not measured/quantified?

On P. 6: please explain role of aphidicolin as done for hydroxyurea

On P. 6/ Fig. 2: I think it would be necessary to have a Western included that shows GFP content after MG132 and roscovitine treatment - not only for Zeocin. It would also be good to have a wild type control here that shows more clearly which band one shall look at. Roscovitine should be introduced/explained here already as it is part of this figure. It becomes confusing as it is somehow handled separately in the following section

On p. 8: 'that ATP-dependent phosphorylation is prerequisite for protein degradation in vitro'. I think the authors should be cautious with such a statement. The results in Supp Fig. 5b do not necessarily support this. Ubiquitination is an ATP dependent process, and lack or insufficient amounts of ATP may simply prevent proteasomal degradation, independently of phosphorylation

On p. 8: 'proteasome-mediated degradation of MYB3R3 is dependent on CDK phosphorylation, and is suppressed by DNA damage'. Such a statement is also very speculative based on in vitro data. Here, the authors need to include some in planta data that show stability in context with genotoxic stress etc. Since GFP-MYB3R3 lines are available, the authors could test for example content of the protein after cycloheximide treatment e.g. in combination with rescovitine and/or zeocin...

On p.9: 'non'phosphorylatable alanine substitutions at all three sites (hereafter called MYB3R3AAA) suggesting that CDK phosphorylation at these residues is indeed responsible for MYB3R3 degradation.' How does the DDD mutant behave here? Is it increasingly instable, or also stable?

On p. 10: '...results suggest that accumulation of the non-phosphorylated form of MYB3R3 is crucial for growth arrest in response to DNA damage '. Please support such a statement by Western-blot to show increased content of the AAA protein, and also with some experiments that show a clear connection to cell cycle. Also, maybe the mutated protein has been rendered non-functional based on the triple change (this may also hold true for the DDD version). It would be good to have here some data that actually show functionality of the protein. Do these MYB3R proteins interact with each other? And if yes, how do the mutated versions behave? How do they bind to target promoters in planta? The mutated forms should have been included in Fig. 5!!

On p. 14: 'we showed that CDKs phosphorylate Rep-MYBs and promote their degradation via the ubiquitin-proteasome pathway'. I agree with the phosphorylation, but I disagree that CDKs indeed promote MYB3Rs degradation in planta. This part is lacking. The authors conclude here mainly from in

vitro data back to in planta. It can even not be excluded that other kinases phosphorylate these MYBs, and cause degradation

Responses to Reviewer #1

Comment:

My main concern with the paper is perhaps a more direct demonstration that the factors described directly affect the G2/M transition. The experiments use "knockout" mutants to demonstrate resistance (or hypersensitivity in the case of r4) to DNA damage. The authors use a range of different damaging agents, but I would have liked to see inclusion of an agent such as camptothecin that more specifically induces DSBs during S-phase. This might be more informative than a general DSB inducing agent or replication inhibitor.

Response:

We examined the response to camptothecin, as shown in Appendix Fig. 1. *myb3r3* and *myb3r5*, as well as *sog1-1*, displayed the same sensitivity to camptothecin as wild-type, indicating that SOG1 and Rep-MYBs are not involved in the response to camptothecin-induced DSBs. This contrasts with their involvement in the response to general DSB-inducing agents (i.e., zeocin, gamma rays, etc.) (Fig. 1b and Supplementary Fig. 4). Camptothecin may provoke a distinct response due to replication fork stalling, but the underlying mechanism remains unknown. In any case, the camptothecin data will not help readers to understand the link to the cell cycle, and we have therefore not included them in the revised manuscript.

Comment:

More importantly, the authors provide nice data regarding induction of G2-specific genes, but in my mind this is indirect evidence for G2 arrest in dividing cells. The authors are careful in their model to state induction of G2/M specific genes, but in the discussion say this leads to G2 arrest. This may likely be the case, but I don't think the experiments directly demonstrate this. Are G2/M genes being turned on but would we see S-phase arrest? Perhaps the authors could show the cellular DNA content using cell sorting of GFP+ cells shortly after DNA damage in their MYB3R3-GFP or MYB3R3AAA-GFP lines? If their assumption is correct, we would expect these cells to be mostly arrested as 4N. If the authors believe this type of experiment is not possible, then they should at least make it more clear in the discussion that direct demonstration of the G2/M arrest still needs to be explored. Or perhaps the authors could provide evidence in the literature to make a stronger case for their assumption.

Response:

Thank you very much for your helpful comments. Cell sorting is a good approach to estimate changes in cell cycle progression. However, it is difficult to isolate dividing 4C cells from endocycling 4C cells, whose numbers increase upon DNA damage (Adachi *et al.*, PNAS 108, 10004–10009, 2011); thus, preparing samples for flow cytometry is tricky. We therefore took a microscopic approach to analyze only meristematic cells (Figure 3). The results showed that zeocin treatment increased the nuclear area, which is known to correlate well with DNA content. In the meristem expressing MYB3R3-GFP with alanine substitutions, nuclei were already enlarged in the absence of zeocin. Our results suggest that DNA damage arrests the cell cycle at late S-to-G2, thereby increasing DNA content

and nuclear size. Since the 4C level of DNA content cannot clearly distinguish G2-phase cells from cells in late S phase, we have left open the possibility that DNA-damaged cells are arrested at late S phase. We have carefully explained this new result in the revised manuscript.

Comment:

Line numbers would be welcome in the manuscript. Also inclusion of figure legends with the figures is helpful.

Response:

We have included line numbers in the revised manuscript. Regarding figure legends, we need to upload the figure files in TIFF format according to the journal's instructions, and the online submission system automatically converts them to PDF files. Thus, we could not include figures legends with the figures. We are sorry for this inconvenience.

Comment:

Do the myb3/5 mutants show the gamma-plantlet phenotype described for the original sog mutant?

Response:

Yes, they do. We have shown the data in Supplementary Fig. 5.

Comment:

Page 3, line ??, last sentence in first paragraph, "(refs 12, 13)" should be standard formatting?

Response:

This is the standard format of *Nature Communications*.

Comment:

Page 3, second paragraph, "that of" can be eliminated, second to last sentence.

Response:

We have eliminated "that of".

Comment:

Page 14, second paragraph, "showed" change to "show".

Response:

We have changed "showed" to "show".

Comment:

Quantification of GFP cells in Figure 2/4 would be helpful to support your arguments of differences in expression. We use confocal microscopy and simply quantify GFP signals using standard software.

Response:

Thank you very much for your useful comment. We have included immunoblot data for MYB3R3-GFP and MYB3R5-GFP. The amount of MYB3R4-GFP protein was too low to detect by immunoblotting, so we have shown quantified GFP fluorescence according to your suggestion (Supplementary Fig. 8).

Comment:

Some of the graphs use symbols in them, but not in the legends. This made it difficult to distinguish the various lines in black and white prints.

Response:

We have made the symbols clearer in each graph. We hope that the various lines are easy to distinguish in the new versions.

Responses to Reviewer #2

Comment:

I have one general query concerning the choice to place a lot of the figures in the supplemental data. I assume this is because of space considerations, but the editor might consider moving some of the Suppl. Figs to the main text. This is not essential, but I feel that the model (Suppl Fig 12) really should be in the article itself.

Response:

We have transferred the supplementary figures depicting the *in vitro* degradation assay and the model to the figures of the article itself. However, since the maximum number of figures that can be included in an article is 10, there still remain 15 supplementary figures.

Comment:

page 2, line 1 - This first sentence needs rephrasing. Perhaps fuse it with the next sentence to give "Inhibition of cell division is an active response to DNA damage..."

Response:

Thank you very much for your correction. We have amended the sentence according to the reviewer's suggestion.

Comment:

page 2, line 16 - It is of course true that environmental influences cause DNA damage, but cellular processes (replication, ROS, ...) also do so. Only mentioning environmental effects is a pity.

Response:

We have also mentioned cellular processes as causes of DNA damage in this sentence.

Comment:

page 4, line 17 -the authors should specify that their work on DSBs and G2 arrest is about Arabidopsis.

Response:

We have indicated that this work was done with *Arabidopsis*.

Comment:

page 4, line 19 - These mutants are properly referenced in the Methods section and have been previously published, but I feel that a short description (more than the word "knockout") would be helpful to readers here.

Response:

We have described the *myb3r* mutants in more detail in the revised manuscript.

Comment:

page 5, line 6. *suppl fig 1a - the myb3r3 and sog1 myb3r3 mutants reach a plateau at 5-6 days. This does not seem to be the case for the sog1 plants? I'm not sure that saying that they are "comparable" is enough here.*

Response:

The previous data showed only a slight difference for *myb3r3* after 6 days of zeocin treatment, which we assumed was caused by variable conditions in measuring root growth on agar plates. We have conducted the measurement again and found that there was no difference between *myb3r3*, *sog1-1* and *myb3r3 sog1-1*. We have shown the data in Supplementary Fig. 2b but have not changed the text.

Comment:

Page 5, Line 9, suppl 1b : a similar effect is visible in the myb3r3 myb3r5 double mutant, but not in the singles (not in myb3r3 in this experiment?). The authors should provide some discussion of this.

Response:

Again, we assumed that a slight difference for *myb3r3 myb3r5* after 6 days of zeocin treatment was caused by variable growth conditions. We repeated the measurement and found that there was no difference between *myb3r3*, *myb3r5* and *myb3r3 myb3r5*. We show the data in Supplementary Fig. 2a and have not changed the text.

Comment:

Page 5, Lines 9-10. The similar phenotypes of the two single and the double mutants shows that both (not either) of them are required and imply epistasis of the two proteins (same pathway).

Response:

Thank you very much for your comment. We have changed the sentence to "both *MYB3R3* and *MYB3R5* are essential for root growth arrest".

Comment:

*Page 5, Lines 10-12 - this sentence is very complicated. Perhaps replace it with: "Observation of the root tip showed that increasing zeocin concentrations resulted in stronger (*myb3r4*) or weaker (*myb3r3*, *myb3r5*) reduction of the meristem size compared to the WT (Fig. 1c; Supplementary Fig. 2). "*

Response:

Our new results showed that the zeocin response of *myb3r4* was the same as that of WT. Therefore, we have amended the description of the *myb3r4* phenotype, including this sentence, in the revised manuscript.

Comment:

Page 5, Line 20 - I suggest replacing "...displayed tolerance..." with "...were significantly more tolerant..."

Response:

We have amended the sentence according to the reviewer's suggestion.

Responses to Reviewer #3

Comment:

CDK targets both Act and Rep-MYB3Rs. How is this different? Both CDKA and CDKB? What is the effect on stability with respect to MYB3R4? In this context, a more complete analysis on R3, R4 and R5 for the majority of the experiments is required (regarding the Rep and Act behavior), e.g. expression pattern, binding to targets, double mutants, etc

Response:

Act-MYBs (e.g., MYB3R4) are phosphorylated and activated by CDK, and are involved in transcriptional activation of G2/M-specific genes under normal growth conditions. However, MYB3R4 does not have a role under DNA stress because the transcript level is rapidly and dramatically reduced by DNA damage (Supplementary Fig. 6). In the revised manuscript, we have further shown that the MYB3R4 protein level was also dramatically reduced by zeocin treatment (Supplementary Fig. 8a, b). Indeed, the *myb3r4* mutant displayed the same zeocin response as WT (Fig. 1). Therefore, in this manuscript, we focused on how the DNA damage response is controlled by Rep-MYBs, and did not intend to report the function or phosphoregulation of Act-MYB (MYB3R4) in the absence of DNA damage. Please note that we showed the transcript and protein levels of MYB3R4 in the presence of MG132 or roscovitine (Supplementary Fig. 8c, d and Supplementary Fig. 9). The results indicated that CDK activity down-regulates the mRNA level, but not the protein level, of MYB3R4 under normal growth conditions.

Comment:

To further connect CDK activity to MYB3R activity, how does, for example, roscovitine impact target gene expression in absence/presence of zeocin?

Response:

Thank you very much for your thoughtful comment. At a high concentration, roscovitine inhibits CDK activities and leads to cell cycle arrest. Even at a lower concentration, it interferes with cell cycle progression, thus perturbing cell cycle-dependent expression of genes including those controlled by MYB3Rs, independently from DNA damage. Therefore, we think that analysing the expression of target genes with a combination of DNA damage and roscovitine would be very tricky, and that the results would be difficult to interpret. For these reasons, we did not attempt this experiment during the revision.

Comment:

*How is the whole cascade connected to CDKA/B on a genetic level? Phenotype of *cdka/b* mutants, the MYB3R phosphostatus and the impact on the target gene expression in the absence presence of A or D variants (complementation)?*

Response:

The homozygous mutant of *CDKA*, encoding the major CDK in *Arabidopsis*, is very rarely obtained, and it exhibits severe growth defects due to impaired cell division (Nowack *et al.*,

Dev. Cell 22, 1030-1040, 2012). Therefore, it would be practically impossible to examine the DNA damage response of the *cdka* mutant. We tried to detect the phosphostatus of MYB3Rs, but it was technically difficult to identify the phosphorylated form of MYB3Rs in protein extracts from wild-type plants. In the revised manuscript, we have included expression analysis of target genes in *myb3r3* expressing *MYB3R3-GFP*, *MYB3R3AAA-GFP* or *MYB3R3DDD-GFP* according to the reviewer's suggestion. The results showed that *MYB3R3-GFP* and *MYB3R3AAA-GFP*, but not *MYB3R3DDD-GFP*, repressed *KNOLLE* and *CYCBI;2* under DNA stress (Fig. 8b). This suggests that CDK phosphorylation of MYB3R3 is associated with target gene expression.

Comment:

I do not fully follow the logic of the A substitution data. To me it seems that S461A has decreased activity. In addition, it seems that S394A is the dominant target of CDKA (based on the double and triple combinations, only activity is lost when S394A is present), but as a single variant there is still activity. This needs to be carefully looked into and better explained. In this context, studying full AAA variants in the context of stability and activity might lead to some misinterpretation, and also the effect of single A and D variants on protein degradation, target gene expression, zeocin impact on stability, etc needs to be investigated. ChIP-PCR should be performed on AAA and DDD variants (also in absence/presence of zeocin).

Response:

Thank you very much for your comment. We have revised the text to explain more clearly the results of phosphorylation assays. We think that S394 is the major phosphorylation site, but T407 and S461 are also targeted by CDK. The persistent phosphorylation level observed for S394A is mysterious, but such a problem often arises when multiple alanine substitutions are introduced into a target protein. One explanation would be that an alanine substitution at one site enhances phosphorylation of another site due to conformational change. Nonetheless, our mass spectrometry analysis showed that all three sites are phosphorylated (Supplementary Fig. 10). This means that, although we do not know the phosphorylation level of each site, all three residues are targeted by CDK. Phosphorylation at multiple sites was also supported by the observation that two conserved domains, one containing S394/T407 and the other S461, were both phosphorylated by CDKA immunoprecipitates (Appendix Fig. 2). We therefore think that introducing A or D substitutions at the three sites is reasonable, and we did not test single A or D variants. Please note that a tobacco Act-MYB, NtmybA2, is phosphorylated by CDK at many sites, and each site makes a contribution to CDK-induced activation (Araki *et al.*, *J. Biol. Chem.* 279, 32979–32988, 2004). In response to the reviewer's last point, ChIP-PCR was conducted on *MYB3R3AAA-GFP* and *MYB3R3DDD-GFP* (Fig. 8a). The results showed that the phosphorylation state of MYB3R3 determines its binding to target promoters.

Comment:

To support the model of competitive activity of (phosphorylated) Rep and Act-MYB3Rs, using ChIP-PCR in respective mutant backgrounds, but also using MYB3R A and D variants (for the importance of phosphorylation) is necessary.

Response:

As mentioned above, MYB3R4 is severely depleted under DNA damage conditions; thus, we assume that MYB3R4 is replaced by Rep-MYBs that accumulate highly in response to DNA damage, but not through competitive activity of Rep- and Act-MYBs. Therefore, in this manuscript, we focus on Rep-MYBs in terms of their involvement in the DNA damage response. ChIP-PCR results with MYB3R3AAA-GFP and MYB3R3DDD-GFP are shown in Fig. 8a in the revised manuscript. We cannot work on ChIP-PCR for MYB3R4 with zeocin-treated samples because only trace amounts of the protein are present.

Comment:

In the discussion it is not clear what has been shown previously (references), what is new here and what is speculation. This section needs to be thoroughly rewritten and focus on those aspects of the model where actual data is available for. The interaction of Rep-MYBs with RBR and E2F was mentioned, but does this depend on the phosphostatus of the Rep-MYBs?

Response:

We have thoroughly revised the first part of the discussion according to the reviewer's suggestion. The involvement of phosphorylation in interactions between MYB3Rs and RBR/E2F has not yet been examined. This is an interesting topic, but further analysis is required to address this point.

Comment:

With respect to the data in Fig 2c and 4a, a comparative quantification (in one experiment) is needed for WT, AAA and DDD.

Response:

We have shown immunoblot data in Fig. 2d, Fig. 6b and Supplementary Fig. 7b.

Comment:

In the figure legends (and/or methods) it should be indicated what $n = 3$ stands for in the context of PCR (biological or technical repeats). One should at least use 3 biological repeats.

Response:

In the figure legends, we have indicated that we used three biological replicates.

Comment:

Regarding the figures (graphs), it might be useful to indicate the % reduction with respect to a phenotype such as root length. In addition, the labels need to be more clearly indicated in color and shape for wt and mutants in the legend. Given the overlap between the various

lines in Figures (such as 4c), it might be useful to also show a bar diagram of 1 time point (with maximal difference).

Response

To show the difference between samples under control conditions, we prefer to display absolute values for both control and zeocin-treated conditions. Regarding the labels, we have made colors and symbols clearer in each graph. We noticed that many lines are overlapping in the new Fig. 6d, and thus have shown a bar diagram for samples of + zeocin, 6 days (Fig. 6e). For the other relevant figures, we think that the original graphs display sufficiently clearly the differences between various lines, and thus have not added bar diagrams.

Comment:

For various blots, the ratio relative to the control should be calculated. Also the loading control on Figure S5 needs to be shown.

Response:

We have indicated the relative intensity of each band for immunoblot and kinase assay. We have also included the loading control (CBB staining) in the new Fig. 4.

Responses to Reviewer #4

Comment:

Reading through the manuscript I find that the authors relate very often to cell cycle, however, most evidence (except for the very last part) is only provided through cell number, and this also only in the very first figure (1c). Personally I think that this should have been done in more elaborated way/more thoroughly to support statements, especially since the authors strongly correlate their findings with cell cycle. Root elongation as it was most often done by the authors is to my opinion not a very good indicator for cell cycle problems.

Response:

Thank you very much for your helpful comments. In the revised manuscript, we have estimated the cell cycle stage of MYB3R3-accumulating cells. Preparation of samples for flow cytometry is tricky due to contamination by endocycling 4C cells, which increase upon DNA damage; thus, we estimated DNA content by measuring the nuclear area of meristematic cells under a microscope (Fig. 3). The results showed that zeocin treatment elevated the nuclear area, and that cells expressing the alanine-substituted version of MYB3R3 have larger nuclei than those expressing wild-type MYB3R3 in the absence of zeocin. Our results suggest that DNA damage arrests the cell cycle at late S-to-G2, thereby increasing DNA content and nuclear size, and that the non-phosphorylated form of MYB3R3 promotes cell cycle arrest.

Comment:

Can the authors explain why they only investigated three out of five myb3r null mutants, and also only a single allele for each of these in this work? I assume that all of these mutants have been backcrossed at least twice? Maybe the authors can add these statements in the method and result part where suitable

Response:

MYB3R2 is a distinct type, and MYB3R1 has only a supplementary function to Act- and Rep-MYBs (Kobayashi *et al.*, *EMBO J.* 34, 1992–2007, 2015). We therefore excluded these two MYB3Rs from further analyses. We have added these statements in the revised manuscript. We have recently noticed that *myb3r4-1* exhibited a different zeocin response compared to the other *myb3r4* alleles; namely, root growth of *myb3r4-1* was slower than wild-type in the presence of zeocin (see Fig. 1b in the last version), while *myb3r4-2* (Haga *et al.*, *Development* 134, 1101–1110, 2007) and GK_634B01 (a new allele isolated from GABI-Kat lines) did not show any difference from wild-type (see Appendix Fig. 3). Therefore, we have decided to use *myb3r4-2* in the revised manuscript. We also tested two alleles for each of *myb3r3* and *myb3r5*, and found that they displayed the same level of zeocin tolerance, as shown in Appendix Fig. 3. Based on these observations, we used *myb3r3-1*, *myb3r4-2* and *myb3r5-1* as *myb3r3*, *myb3r4* and *myb3r5*, respectively, in the revised manuscript. Since all of these are published alleles (Kobayashi *et al.*, 2015; Haga *et al.*, 2007), we have just added brief statements in the first paragraph of the Methods section.

Comment:

On P. 4: I would disagree with the statements 'Zeocin tolerance was comparable between myb3r3, sog1-1 and the myb3r3 sog1-1' based on what is show in Supp. Fig. 1a, day 6. myb3R3 looks more sensitive compared to sog1-1 or the double. Likewise, I would disagree with the statement 'The myb3r3 myb3r5 double mutant was tolerant to zeocin to an extent similar to each single mutant', the graph in Supp Fig. 1b at day 6 does not support this statement.

Response:

The data in the previous version of the manuscript showed only a slight difference after 6 days of zeocin treatment, and we assumed that this was caused by variable conditions in measuring root growth on agar plates. We have conducted the measurement again and found that there was no difference between *myb3r3*, *sog1-1* and *myb3r3 sog1-1*, or between *myb3r3*, *myb3r5* and *myb3r3 myb3r5*. We have shown the new data in Supplementary Fig. 2 and have not changed the text.

Comment:

Furthermore, why did the authors only generate a sog1-1myb3r3 double? Why has this not done for the others? How do these mutants behave in a dose-dependent experiments trying different concentrations of Zeocin?

Response:

Since *myb3r4* showed the same zeocin response as wild-type, we focused on Rep-MYBs. Moreover, since the *myb3r3 myb3r5* double mutant showed the same zeocin tolerance as that of each single mutant, we only generated the *myb3r3 sog1-1* double mutant. In the revised manuscript, we have first explained the data for *myb3r3 myb3r5*, and then for *myb3r3 sog1-1*, to facilitate understanding (Supplementary Fig. 2). We have also added the root growth data from dose-dependent experiments using 0 to 4 μM zeocin (Supplementary Fig. 1).

Comment:

On P. 5 the authors state 'Observation of the root tip showed that increasing zeocin concentration resulted in stronger or weaker reduction of the meristem size in myb3r4 or myb3r3'. Can the authors refer here to cell number rather than size, as the latter was not measured/quantified?

Response:

Thank you very much for your comment. We have amended the sentence according to the reviewer's suggestion.

Comment:

On P. 6: please explain role of aphidicolin as done for hydroxyurea

Response:

In the original manuscript, we mentioned that aphidicolin is a drug which inhibits DNA polymerase α . We think this is sufficient to indicate the role of aphidicolin, and have retained the original sentence in the revised manuscript.

Comment:

On P. 6/ Fig. 2: I think it would be necessary to have a Western included that shows GFP content after MG132 and roscovitine treatment - not only for Zeocin. It would also be good to have a wild type control here that shows more clearly which band one shall look at.

Response

We have added the immunoblot data in the revised manuscript (Fig. 2d). We have included the *myb3r3* mutant, rather than wild-type, as a control because *ProMYB3R3::MYB3R3-GFP* had been introduced into *myb3r3*.

Comment:

Roscovitine should be introduced/explained here already as it is part of this figure. It becomes confusing as it is somehow handled separately in the following section

Response:

It is necessary to explain why we examined the possibility of CDK phosphorylation, and so we prefer to describe the roscovitine data in a different section, in which two papers describing Act-MYB phosphorylation are first introduced. As a result, the data in Fig. 2c and d are explained in two separate sections, but we think that it is not so confusing.

Comment:

On p. 8: 'that ATP-dependent phosphorylation is prerequisite for protein degradation in vitro'. I think the authors should be cautious with such a statement. The results in Supp Fig. 5b do not necessarily support this. Ubiquitination is an ATP dependent process, and lack or insufficient amounts of ATP may simply prevent proteasomal degradation, independently of phosphorylation

Response:

Thank you very much for your helpful comment. We have amended the sentence according to the reviewer's suggested qualification.

Comment:

On p. 8: 'proteasome-mediated degradation of MYB3R3 is dependent on CDK phosphorylation, and is suppressed by DNA damage'. Such a statement is also very speculative based on in vitro data.

Response:

This statement was based not only on *in vitro* data but also on *in planta* data shown in Fig. 2. We have clarified this point in the revised manuscript.

Comment:

Here, the authors need to include some in planta data that show stability in context with genotoxic stress etc. Since GFP-MYB3R3 lines are available, the authors could test for example content of the protein after cycloheximide treatment e.g. in combination with roscovitine and/or zeocin...

Response:

Thank you very much for your thoughtful comments. Cycloheximide treatment inhibits DNA damage-induced accumulation of CDK inhibitors, which is controlled by the transcription factor SOG1 (Yi *et al.*, *Plant Cell* 26, 296–309, 2014), thus perturbing the DNA damage response. Therefore, we think that simultaneous application of cycloheximide and zeocin is not reasonable. One approach might be to treat first with zeocin (or roscovitine) and then with cycloheximide. However, since the amount of MYB3R3-GFP protein after zeocin (or roscovitine) treatment is considerably higher than the control without drug treatment, it is tricky to compare the degradation rate after cycloheximide treatment with that of the control. We understand that the present data do not directly address the protein stability, but on the other hand, our qRT-PCR and immunoblotting data demonstrate higher protein accumulation of Rep-MYBs upon zeocin or roscovitine treatment. In the revised manuscript, therefore, we have carefully discussed protein accumulation, rather than protein stability, of Rep-MYBs. We noticed that we did not show *MYB3R* transcript levels after roscovitine treatment, and we have thus added these data in Supplementary Fig. 9.

Comment:

On p.9: 'non'phosphorylatable alanine substitutions at all three sites (hereafter called MYB3R3AAA) suggesting that CDK phosphorylation at these residues is indeed responsible for MYB3R3 degradation.' How does the DDD mutant behave here? Is it increasingly instable, or also stable?

Response:

We did not test MYB3R3DDD in the *in vitro* degradation assay. However, please note that, regardless of the presence of zeocin, MYB3R3DDD-GFP accumulated to a lesser extent than MYB3R3-GFP, suggesting its unstable nature (Fig. 6a, b).

Comment:

On p. 10: '...results suggest that accumulation of the non-phosphorylated form of MYB3R3 is crucial for growth arrest in response to DNA damage '. Please support such a statement by Western-blot to show increased content of the AAA protein, and also with some experiments that show a clear connection to cell cycle. Also, maybe the mutated protein has been rendered non-functional based on the triple change (this may also hold true for the DDD version). It would be good to have here some data that actually show functionality of the protein. Do these MYB3R proteins interact with each other? And if yes, how do the mutated versions behave? How do they bind to target promoters in planta? The mutated forms should have been included in Fig. 5!!

Response:

Thank you very much for your helpful comments. In the revised manuscript, we have added the data of immunoblotting (Fig. 6b) and nuclear area of MYB3R3-accumulating cells (Fig. 3), the latter of which suggests that the non-phosphorylated form of MYB3R3 promotes cell cycle arrest at late S/G2. The matter of functionality of mutated proteins is difficult to resolve because the protein level also changes when mutations are introduced. The root growth data and the ChIP-PCR result indicate that MYB3R3AAA is functional (Fig. 6d, e and Fig. 8), but we cannot deny the possibility that MYB3R3DDD has lower functionality than MYB3R3. Therefore, we have carefully discussed these results in the revised manuscript. It remains unknown whether MYB3R proteins interact with each other, and so we have not discussed this matter. We have added the ChIP-PCR data for MYB3R3AAA and MYB3R3DDD in Fig. 8a.

Comment:

On p. 14: 'we showed that CDKs phosphorylate Rep-MYBs and promote their degradation via the ubiquitin-proteasome pathway'. I agree with the phosphorylation, but I disagree that CDKs indeed promote MYB3Rs degradation in planta. This part is lacking. The authors conclude here mainly from in vitro data back to in planta. It can even not be excluded that other kinases phosphorylate these MYBs, and cause degradation

Response:

We think that the roscovitine data suggest the role of CDK activity in controlling protein accumulation of Rep-MYBs. Moreover, higher and lower accumulation of MYB3R3AAA and MYB3R3DDD, respectively, also suggest that CDK phosphorylation is involved in the control of Rep-MYB accumulation. In the revised manuscript, we have carefully discussed the involvement of CDK activity in the control of Rep-MYB accumulation.

Reviewers' comments:

Reviewer #1 (Remarks to the Author):

The points raised in the review have been addressed but not fully resolved. However, I do (somewhat) agree that the points raised could be beyond the scope of this manuscript.

Reviewer #2 (Remarks to the Author):

I am satisfied with the responses of the authors to my comments. There is one point, that comes up from this however, which I don't feel affects the interest of the manuscript, just implies some minor editing to avoid overinterpretation of the results from the myb3r4 mutant.

In response to my query, the revised manuscript includes a brief description of the mutant alleles which notes that the r4 allele is a promoter mutation causing transcriptional down regulation of the gene, while r3 and r5 mutant alleles are knockouts. Given the essentially WT behaviour of the r4 mutant and the presence of ~3% native MYB3r4 transcript, this means that great care needs to be taken to avoid ambiguity in any conclusions made with this mutant. Specifically:

- p5 - the fact that "...Root growth in myb3r4 was inhibited by zeocin to the same extent as in WT (Fig. 1b)" can not be used to conclude "... that this Act-MYB is not involved in suppressing root growth in response to DSBs." Maybe 3% transcript is enough in this context (but of course maybe not in all contexts).

- on the next page, this argument also makes sense of the fact that "increasing zeocin concentrations resulted in weaker reduction of the meristem cell number in myb3r3 and myb3r5, but not in myb3r4, compared to WT" and of "dead cells were observed in stem cells and stele precursor cells of WT and myb3r4 in the presence of 1 μ M zeocin, while a higher concentration was required for myb3r3 and myb3r5".

In the methods section the authors mention that they verified the Bleomycin sensitivity with another allele (is this a KO?), and in any case the myb3r4 mutant is a very minor part of this work. As I say above, this is a minor point and I feel that it can be dealt with by minor editing changes to avoid ambiguity.

Reviewer #3 (Remarks to the Author):

This revised manuscript has improved greatly, but I still have a few remaining considerations:

* All genetic analyses are done on one allele for myb3r3 or myb3r5. Can the authors include some assays on another allele?

* All the phosphorylation site data are based on in vitro analyses. Can the authors include an identification of in vivo sites (e.g. transiently in tobacco, or in 35S::CDKA, ...)?

* Phosphorylation seems to control protein accumulation, but this is misleading (suggests promoter binding) in the Figure 10 model.

* Regarding Figures 1b-c, it seems that this is a complete repeat from an earlier experiment that is depicted, but the previous trends are not maintained. What is the cause for this?

* How are the ratio's in the blots calculated? The whole region? The band? This should obviously be corrected for band size and intensity.

Reviewer #4 (Remarks to the Author):

It looks to me that the authors overall went thoroughly through the criticism and addressed most of my criticism satisfactorily. Thank you! One main concern still remains open in that the cell cycle data are relatively poorly analyzed to my opinion. The authors now use cell nucleus size as an indicator for certain phases. I agree that an approach like e.g. FACS might be difficult but it might provide likely more solid data. It is feasible though and with the correct markers introduced may generate more robust information. Another system such as synchronized BY-2 cells would have also been acceptable. Anyway, I think the paper has greatly improved and is convincing to me in its current state.

Responses to Reviewer #2

Comment:

In response to my query, the revised manuscript includes a brief description of the mutant alleles which notes that the r4 allele is a promoter mutation causing transcriptional down regulation of the gene, while r3 and r5 mutant alleles are knockouts. Given the essentially WT behaviour of the r4 mutant and the presence of ~3% native MYB3r4 transcript, this means that great care needs to be taken to avoid ambiguity in any conclusions made with this mutant. Specifically:

- p5 - the fact that "...Root growth in myb3r4 was inhibited by zeocin to the same extent as in WT (Fig. 1b)" can not be used to conclude "... that this Act-MYB is not involved in suppressing root growth in response to DSBs." Maybe 3% transcript is enough in this context (but of course maybe not in all contexts).

- on the next page, this argument also makes sense of the fact that "increasing zeocin concentrations resulted in weaker reduction of the meristem cell number in myb3r3 and myb3r5, but not in myb3r4, compared to WT" and of "dead cells were observed in stem cells and stele precursor cells of WT and myb3r4 in the presence of 1 μ M zeocin, while a higher concentration was required for myb3r3 and myb3r5".

In the methods section the authors mention that they verified the Bleomycin sensitivity with another allele (is this a KO?), and in any case the myb3r4 mutant is a very minor part of this work. As I say above, this is a minor point and I feel that it can be dealt with by minor editing changes to avoid ambiguity.

Response:

Thank you very much for your comment. In the revised manuscript, we have included the data for another *myb3r4* allele, *myb3r4-3*, which produces no *MYB3R4* transcripts (Supplementary Fig. 4). Our measurement of root growth showed that this mutant exhibited the same sensitivity to zeocin as wild-type and *myb3r4-2*, the latter of which produces < 3% transcripts (Supplementary Fig. 2). This indicates that *MYB3R4* is not involved in controlling cell division in response to DNA damage. We have explained this new result in the second paragraph of the Results section.

Responses to Reviewer #3

Comment:

All genetic analyses are done on one allele for myb3r3 or myb3r5. Can the authors include some assays on another allele?

Response:

We have included the root growth data for a second allele for each *MYB3R* (Supplementary Fig. 2). We obtained similar data to the original alleles.

Comment:

All the phosphorylation site data are based on in vitro analyses. Can the authors include an identification of in vivo sites (e.g. transiently in tobacco, or in 35S::CDKA, ...)?

Response:

Since the protein levels of Rep-MYBs are very low in the absence of DNA damage, we attempted to overexpress MYB3R3 to obtain sufficient amounts of protein for MS analysis. First, we tried to overexpress MYB3R3-YFP under the 35S promoter in *Arabidopsis* plants. However, we could not detect intense bands corresponding to MYB3R3-YFP on immunoblots (Appendix Figure, slides #1 and #2). We were also unable to overexpress Venus-fused MYB3R3 in protoplasts of either *Arabidopsis* or tobacco cultured cells (Appendix Figure, slides # 1-3; note that the immunoblots showed many bands because we used a high-sensitivity kit for immunodetection). Therefore, we gave up trying to collect phosphorylated MYB3R3 from plant cells for MS analysis. We think that phosphorylated Rep-MYBs must be highly unstable *in vivo*.

Comment:

Phosphorylation seems to control protein accumulation, but this is misleading (suggests promoter binding) in the Figure 10 model.

Response:

Thank you very much for your comment. We have revised Fig. 10 to remove this ambiguity.

Comment:

Regarding Figures 1b-c, it seems that this is a complete repeat from an earlier experiment that is depicted, but the previous trends are not maintained. What is the cause for this?

Response:

In the original manuscript, we used the *myb3r4* allele *myb3r4-1*. However, during the last revision, we found that *myb3r4-1* exhibited a different zeocin response compared to the other alleles, *myb3r4-2* and *myb3r4-3*; namely, root growth of *myb3r4-1* was slower than wild-type in the presence of zeocin, while *myb3r4-2* and *myb3r4-3* did not show any difference from wild-type. Therefore, we decided to use the data for *myb3r4-2* instead of *myb3r4-1*. This is the reason why the previous trend was not maintained in the last revision. In the current version, we have included the root growth data for both *myb3r4-2* and *myb3r4-3* in response to the comment of reviewer #2 (Supplementary Fig. 2).

Comment:

How are the ratio's in the blots calculated? The whole region? The band? This should obviously be corrected for band size and intensity.

Response:

We measured the intensity in the area of each labelled band, and relative values were calculated by normalizing with respect to the bands of controls. We have indicated this point in the Methods section.

Response to Reviewer #4

Comment:

It looks to me that the authors overall went thoroughly through the criticism and addressed most of my criticism satisfactorily. Thank you! One main concern still remains open in that the cell cycle data are relatively poorly analyzed to my opinion. The authors now use cell nucleus size as an indicator for certain phases. I agree that an approach like e.g. FACS might be difficult but it might provide likely more solid data. It is feasible though and with the correct markers introduced may generate more robust information. Another system such as synchronized BY-2 cells would have also been acceptable. Anyway, I think the paper has greatly improved and is convincing to me in its current state.

Response:

As described in the manuscript, expression of most mitotic genes is repressed upon DNA damage, and the G2/M marker, *CYCBI;1*, is responsive to DNA damage. Therefore, we could not find any appropriate marker that can monitor DNA damage-induced S/G2 arrest. Moreover, BY-2 cells are unsuitable for studies on the DNA damage response, because they are highly tolerant to DNA-damaging agents (our unpublished results). Instead, we previously used *Arabidopsis* cultured cells that are sensitive to zeocin; our data showed that double-strand breaks triggered cell cycle arrest at G2, but not G1 (Adachi *et al.*, 2011). To provide further support for the S/G2 arrest, we performed an additional experiment using EdU labelling (Supplementary Fig. 11). The results showed that cell cycle progression through G2 was delayed and suppressed in the presence of zeocin, and thus that entry to M phase was inhibited. These data give more robust information supporting DNA damage-induced S/G2 arrest; thus, we have included this new result in the revised manuscript. Please note that we have also added Fig. 3c, showing that the nuclear area was increased in root meristematic cells after zeocin treatment. This indicates that G1 arrest does not occur in response to DNA damage.

REVIEWERS' COMMENTS:

Reviewer #3 (Remarks to the Author):

While there are still some issues (different responses of different alleles - but at least 2 seem to support the effect, and no in vivo data for phosphosites), the authors have largely addressed my remaining comments. Regarding the lack of in vivo data, I would encourage the authors to add their attempts and their conclusion to the manuscript.

Response to Reviewer #3

Comment:

While there are still some issues (different responses of different alleles - but at least 2 seem to support the effect, and no in vivo data for phosphosites), the authors have largely addressed my remaining comments. Regarding the lack of in vivo data, I would encourage the authors to add their attempts and their conclusion to the manuscript.

Response:

We have mentioned our attempt to identify *in vivo* phosphorylation sites in the Results section (page 13).